# On the Over-Memorization During Natural, Robust and Catastrophic Overfitting

**Runqi Lin**
Sydney AI Centre, The University of Sydney
rlin0511@sydney.edu.au

**Chaojian Yu**
Sydney AI Centre, The University of Sydney
chyu8051@sydney.edu.au

**Bo Han**
Hong Kong Baptist University
bhanml@comp.hkbu.edu.hk

**Tongliang Liu**\*
Sydney AI Centre, The University of Sydney
tongliang.liu@sydney.edu.au

## Abstract

Overfitting negatively impacts the generalization ability of deep neural networks (DNNs) in both natural and adversarial training. Existing methods struggle to consistently address different types of overfitting, typically designing strategies that focus separately on either natural or adversarial patterns. In this work, we adopt a unified perspective by solely focusing on natural patterns to explore different types of overfitting. Specifically, we examine the memorization effect in DNNs and reveal a shared behaviour termed over-memorization, which impairs their generalization capacity. This behaviour manifests as DNNs suddenly becoming high-confidence in predicting certain training patterns and retaining a persistent memory for them. Furthermore, when DNNs over-memorize an adversarial pattern, they tend to simultaneously exhibit high-confidence prediction for the corresponding natural pattern. These findings motivate us to holistically mitigate different types of overfitting by hindering the DNNs from over-memorization training patterns. To this end, we propose a general framework, *Distraction Over-Memorization* (DOM), which explicitly prevents over-memorization by either removing or augmenting the high-confidence natural patterns. Extensive experiments demonstrate the effectiveness of our proposed method in mitigating overfitting across various training paradigms. Our implementation can be found at https://github.com/tmllab/2024_ICLR_DOM.

## 1 Introduction

In recent years, deep neural networks (DNNs) have achieved remarkable success in pattern recognition tasks. However, overfitting, a widespread and critical issue, substantially impacts the generalization ability of DNNs. This phenomenon manifests as DNNs achieving exceptional performance on training patterns, but showing suboptimal representation ability with unseen patterns.

Different types of overfitting have been identified in various training paradigms, including natural overfitting (NO) in natural training (NT), as well as robust overfitting (RO) and catastrophic overfitting (CO) in multi-step and single-step adversarial training (AT). NO (Dietterich, 1995) presents as the model's generalization gap between the training and test patterns. On the other hand, RO (Rice et al., 2020) is characterized by a gradual degradation in the model's test robustness as training progresses. Besides, CO (Wong et al., 2019) appears as the model's robustness against multi-step adversarial attacks suddenly plummets from a peak to nearly 0%.

In addition to each type of overfitting having unique manifestations, previous research (Rice et al., 2020; Andriushchenko & Flammarion, 2020) suggests that directly transferring remedies from one type of overfitting to another typically results in limited or even ineffective outcomes. Consequently, most existing methods are specifically designed to handle each overfitting type based on characteristics

---

\*Corresponding author

associated with natural or adversarial patterns. Despite the significant progress in individually addressing NO, RO and CO, a common understanding and solution for them remain unexplored.

In this study, we take a unified perspective, solely concentrating on natural patterns, to link overfitting in various training paradigms. More specifically, we investigate the DNNs' memorization effect concerning each training pattern and reveal a shared behaviour termed over-memorization. This behaviour manifests as the model suddenly exhibits high-confidence in predicting certain training (natural or adversarial) patterns, which subsequently hinders the DNNs' generalization capabilities. Additionally, the model persistent a strong memory for these over-memorization patterns, retaining the ability to predict them with high-confidence, even after they've been removed from the training process. Furthermore, we investigate the DNNs' prediction between natural and adversarial patterns within a single sample and find that the model exhibits a similar memory tendency in over-memorization samples. This tendency manifests as, when the model over-memorizes certain adversarial patterns, it will simultaneously display high-confidence predictions for the corresponding natural patterns. Leveraging this tendency, we are able to reliably and consistently identify over-memorization samples by solely examining the prediction confidence on natural patterns, regardless of the training paradigm.

Building on this shared behaviour, we aim to holistically mitigate different types of overfitting by hindering the model from over-memorization training patterns. To achieve this goal, we propose a general framework named *Distraction Over-Memorization* (DOM), that either removes or applies data augmentation to the high-confidence natural patterns. This strategy is intuitively designed to weaken the model's confidence in over-memorization patterns, thereby reducing its reliance on them. Extensive experiments demonstrate the effectiveness of our proposed method in alleviating overfitting across various training paradigms. Our major contributions are summarized as follows:

- We reveal a shared behaviour, over-memorization, across different types of overfitting: DNNs tend to exhibit sudden high-confidence predictions and maintain persistent memory for certain training patterns, which results in a decrease in generalization ability.

- We discovered that the model shows a similar memory tendency in over-memorization samples: when DNNs over-memorize certain adversarial patterns, they tend to simultaneously exhibit high-confidence in predicting the corresponding natural patterns.

- Based on these insights, we propose a general framework DOM to alleviate overfitting by explicitly preventing over-memorization. We evaluate the effectiveness of our method with various training paradigms, baselines, datasets and network architectures, demonstrating that our proposed method can consistently mitigate different types of overfitting.

## 2 RELATED WORK

### 2.1 MEMORIZATION EFFECT

Since Zhang et al. (2021) observed that DNNs have the capacity to memorize training patterns with random labels, a line of work has demonstrated the benefits of memorization in improving generalization ability (Neyshabur et al., 2017; Novak et al., 2018; Feldman, 2020; Yuan et al., 2023). The memorization effect (Arpit et al., 2017; Bai et al., 2021; Xia et al., 2021; 2023; Lin et al., 2022; 2023b) indicates that the DNNs prioritize learning patterns rather than brute-force memorization. In the context of multi-step AT, Dong et al. (2021) suggests that the cause of RO can be attributed to the model's memorization of one-hot labels. However, the prior studies that adopt a unified perspective to understand overfitting across various training paradigms are notably scarce.

### 2.2 NATURAL OVERFITTING

NO (Dietterich, 1995) is typically shown as the disparity in the model's performance between training and test patterns. To address this issue, two fundamental approaches, data augmentation and regularization, are widely employed. Data augmentation artificially expands the training dataset by applying transformations to the original patterns, such as Cutout (DeVries & Taylor, 2017), Mixup (Zhang et al., 2018), AutoAugment (Cubuk et al., 2018) and RandomErasing (Zhong et al., 2020). On the other hand, regularization methods introduce explicit constraints on the DNNs to mitigate NO, including dropout (Wan et al., 2013; Ba & Frey, 2013; Srivastava et al., 2014), stochastic weight averaging (Izmailov et al., 2018), and stochastic pooling (Zeiler & Fergus, 2013).

## 2.3 ROBUST AND CATASTROPHIC OVERFITTING

DNNs are known to be vulnerable to adversarial attacks (Szegedy et al., 2014), and AT has been demonstrated to be the most effective defence method (Athalye et al., 2018; Zhou et al., 2022). AT is generally formulated as a min-max optimization problem (Madry et al., 2018; Croce et al., 2022). The inner maximization problem tries to generate the strongest adversarial examples to maximize the loss, and the outer minimization problem tries to optimize the network to minimize the loss on adversarial examples, which can be formalized as follows:

$$\min_\theta \mathbb{E}_{(x,y)\sim\mathcal{D}} \left[ \max_{\delta\in\Delta} \ell(x+\delta, y; \theta) \right], \tag{1}$$

where $(x, y)$ is the training dataset from the distribution $D$, $\ell(x, y; \theta)$ is the loss function parameterized by $\theta$, $\delta$ is the perturbation confined within the boundary $\epsilon$ shown as: $\Delta = \{\delta : \|\delta\|_p \le \epsilon\}$.

For multi-step and single-step AT, PGD (Madry et al., 2018) and RS-FGSM (Wong et al., 2019) are the prevailing methods used to generate adversarial perturbations, where the $\Pi$ denotes the projection:

$$\eta = \text{Uniform}(-\epsilon, \epsilon),$$
$$\delta_{PGD}^T = \Pi_{[-\epsilon,\epsilon]}[\eta + \alpha \cdot \text{sign}\left(\nabla_{x+\eta+\delta^{T-1}}\ell(x+\eta+\delta^{T-1}, y; \theta)\right)], \tag{2}$$
$$\delta_{RS-FGSM} = \Pi_{[-\epsilon,\epsilon]}[\eta + \alpha \cdot \text{sign}\left(\nabla_{x+\eta}\ell(x+\eta, y; \theta)\right)].$$

With the focus on DNNs' robustness, overfitting has also been observed in AT. An overfitting phenomenon known as RO (Rice et al., 2020) has been identified in multi-step AT, which manifests as a gradual degradation in the model's test robustness with further training. Further investigation found that the conventional remedies for NO have minimal effect on RO (Rice et al., 2020). As a result, a lot of work attempts to explain and mitigate RO based on its unique characteristics. For example, some research suggests generating additional adversarial patterns (Carmon et al., 2019; Gowal et al., 2020), while others propose techniques such as adversarial label smoothing (Chen et al., 2021; Dong et al., 2021) and adversarial weight perturbation (Wu et al., 2020; Yu et al., 2022a;b). Meanwhile, another type of overfitting termed CO (Wong et al., 2019) has been identified in single-step AT, characterized by the model's robustness against multi-step adversarial attacks will abruptly drop from peak to nearly 0%. Recently studies have shown that current approaches for addressing NO and RO are insufficient for mitigating CO (Andriushchenko & Flammarion, 2020; Sriramanan et al., 2021). To eliminate this strange phenomenon, several approaches have been proposed, including constraining the weight updates (Golgooni et al., 2023; Huang et al., 2023a) and smoothing the adversarial loss surface (Andriushchenko & Flammarion, 2020; Sriramanan et al., 2021; Lin et al., 2023a).

Although the aforementioned methods can effectively address NO, RO and CO separately, the understanding and solutions for these overfitting types remain isolated from each other. This study reveals a shared DNN behaviour termed over-memorization. Based on this finding, we propose the general framework DOM aiming to holistically address overfitting across various training paradigms.

## 3 UNDERSTANDING OVERFITTING IN VARIOUS TRAINING PARADIGMS

In this section, we examine the model's memorization effect on each training pattern. We observe that when the model suddenly becomes high-confidence predictions in certain training patterns, its generalization ability declines, which we term as over-memorization (Section 3.1). Furthermore, we notice that over-memorization also occurs in adversarial training, manifested by the DNNs simultaneously becoming high-confidence in predicting both natural and adversarial patterns within a single sample (Section 3.2). To this end, we propose a general framework *Distraction Over-Memorization* (DOM) to holistically mitigate different types of overfitting by preventing over-memorization (Section 3.3). The detailed experiment settings can be found in Appendix A.

### 3.1 OVER-MEMORIZATION IN NATURAL TRAINING

To begin, we explore the natural overfitting (NO) by investigating the model's memorization effect. As illustrated in Figure 1 (left), we can observe that shortly after the first learning rate decay (150th epoch), the model occurs NO, resulting in a 5% performance gap between training and test patterns.

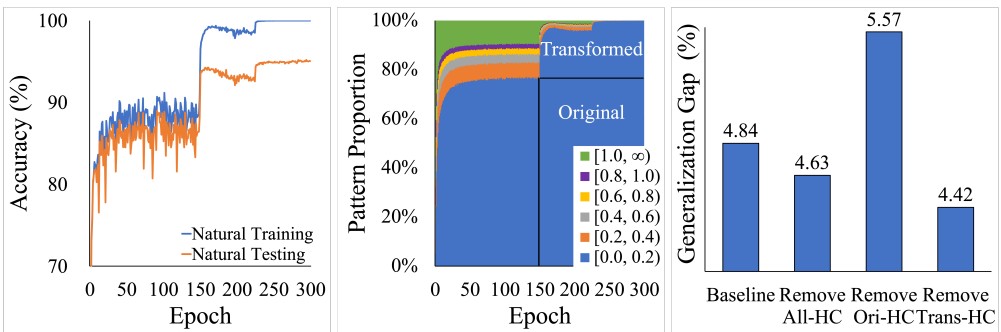

Figure 1. Left Panel: The training and test accuracy of natural training. Middle Panel: Proportion of training patterns based on varying loss ranges. Right Panel: Model's generalization gap after removing different categories of high-confidence (HC) patterns.

Then, we conduct a statistical analysis of the model's training loss on each training pattern, as depicted in Figure 1 (middle). We observe that aligned with the onset of NO, the proportion of the model's high-confidence (loss range 0-0.2) prediction patterns suddenly increases by 20%. This observation prompted us to consider whether the decrease in DNNs' generalization ability is linked to the increase in high-confidence training patterns. To explore the connection between high-confidence patterns and NO, we directly removed these patterns (All-HC) from the training process after the first learning rate decay. As shown in Figure 1 (right), there is a noticeable improvement (4%) in the model's generalization capability, with the generalization gap shrinking from 4.84% to 4.63%. This finding indicates that continuous learning on these high-confidence patterns may not only fail to improve but could actually diminish the model's generalization ability.

To further delve into the impact of high-confidence patterns on model generalization, we divide them into two categories: the "original" that displays small-loss before NO, and the "transformed" that becomes small-loss after NO. Next, we separately remove these two categories to investigate their individual influence, as shown in Figure 1 (right). We can observe that only removing the original high-confidence (Ori-HC) patterns negatively affects the model's generalization (5.57%), whereas only removing the transformed high-confidence (Trans-HC) patterns can effectively alleviate NO (4.42%). Therefore, the primary decline in the model's generalization can be attributed to the learning of these transformed high-confidence patterns. Additionally, we note that the model exhibits an uncommon memory capacity for transformed high-confidence patterns, as illustrated in Figure 2. Our analysis suggests that, compared to the original ones, DNNs show a notably persistent memory for these transformed high-confidence patterns. This uncommon memory is evidenced by a barely increase (0.01) in training loss after their removal from the training process. Building on these findings, we term this behaviour as over-memorization, characterized by DNNs suddenly becoming high-confidence predictions and retaining a persistent memory for certain training patterns, which weakens their generalization ability.

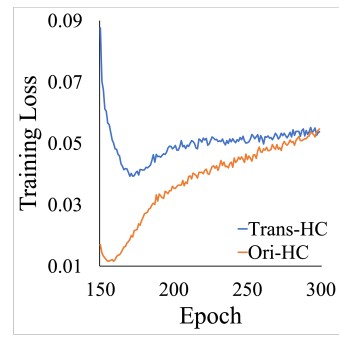

Figure 2. The loss curves for both original and transformed high-confidence (HC) patterns after removing all HC patterns.

## 3.2 OVER-MEMORIZATION IN ADVERSARIAL TRAINING

In this section, we explore the over-memorization behaviour in robust overfitting (RO) and catastrophic overfitting (CO). During both multi-step and single-step adversarial training (AT), we notice that similar to NO, the model abruptly becomes high-confidence in predicting certain adversarial patterns with the onset of RO and CO, as illustrated in Figure 3 (1st and 2nd). Meanwhile, directly removing these high-confidence adversarial patterns can effectively mitigate RO and CO, as detailed in Section 4.2. Therefore, the combined observations suggest a shared behaviour that the over-memorization of certain training patterns impairs the generalization capabilities of DNNs.

Besides, most of the current research on RO and CO primarily focuses on the perspective of adversarial patterns. In this study, we investigate the AT-trained model's memorization effect on natural patterns,

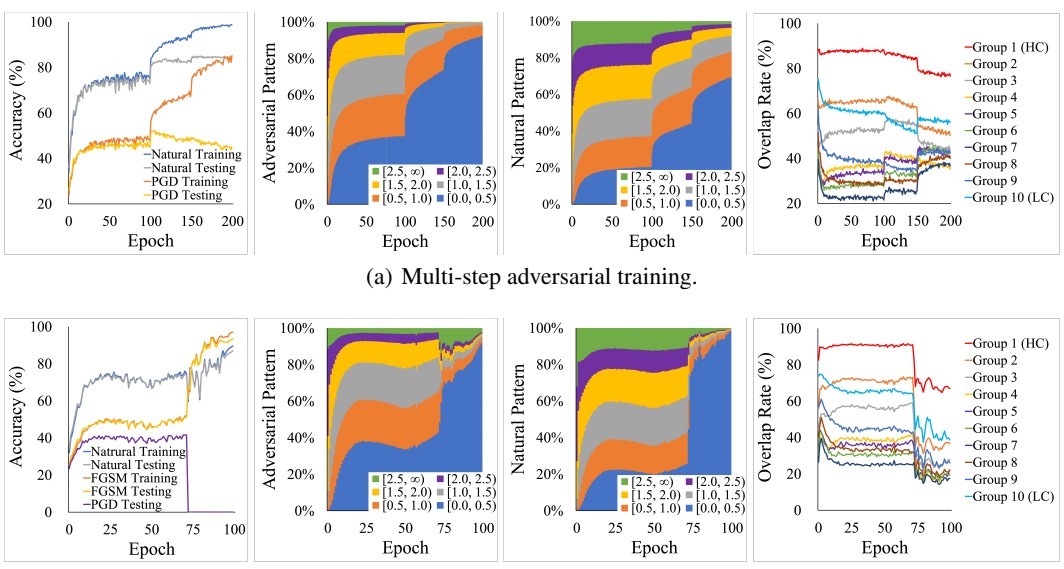

(a) Multi-step adversarial training.

(b) Single-step adversarial training.

Figure 3. 1st Panel: The training and test accuracy of adversarial training. 2nd/3rd Panel: Proportion of adversarial/natural patterns based on varying training loss ranges. 4th Panel: The overlap rate between natural and adversarial patterns grouped by training loss rankings.

as illustrated in Figure 3 (3rd). With the onset of RO and CO, we observe a sudden surge in high-confidence prediction natural patterns within the AT-trained model, similar to the trend seen in adversarial patterns. Intriguingly, the AT-trained model never actually encounters natural patterns, it only interacts with the adversarial patterns generated from them. Building on this observation, we hypothesize that the DNNs' memory tendency is similar between the natural and adversarial pattern for a given sample. To validate this hypothesis, we ranked the natural patterns by their natural training loss (from high-confidence to low-confidence), and subsequently divided them into ten groups, each containing 10% of the total training patterns. Using the same approach, we classify the adversarial patterns into ten groups based on the adversarial training loss as the ranking criterion.

From Figure 3 (4th), we can observe a significantly high overlap rate (90%) between the high-confidence predicted natural and adversarial patterns. This observation suggests that when the model over-memorizes an adversarial pattern, it tends to simultaneously exhibit high-confidence in predicting the corresponding natural pattern. We also conduct the same experiment in TRADES (Zhang et al., 2019), which encounters natural patterns during the training process, and reaches the same observation, as shown in Appendix B. To further validate this similar memory tendency, we attempt to detect the high-confidence adversarial pattern solely based on their corresponding natural training loss. From Figure 4, we are able to clearly distinguish the high-confidence and low-confidence adversarial patterns by classifying their natural training loss. Therefore, by leveraging this tendency, we can reliably and consistently identify the over-memorization pattern by exclusively focusing on the natural training loss, regardless of the training paradigm.

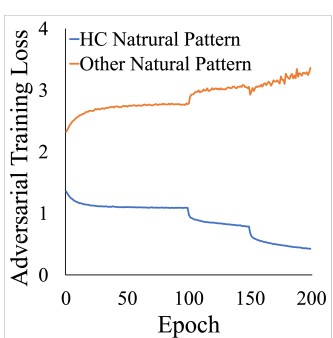

Figure 4. The average loss of adversarial pattern grouped by natural training loss.

### 3.3 PROPOSED APPROACH

Building on the above findings, we propose a general framework, named *Distraction Over-Memorization* (DOM), which is designed to proactively prevent the model from over-memorization training patterns, thereby eliminating different types of overfitting. Specifically, we first establish a fixed loss threshold to identify over-memorization patterns. Importantly, regardless of the training paradigm, DOM exclusively compares the natural training loss with this established threshold. Subsequently, our framework employs two mainstream operations to validate our perspective: removal and

---

**Algorithm 1:** *Distraction Over-Memorization* (DOM)

---

**Input**: Network $f_\theta$, epochs E, mini-batch M, loss threshold $\mathcal{T}$, warm-up epoch $\mathcal{K}$, data argumentation operate $\mathcal{DA}$, data argumentation strength $\beta$, data argumentation iteration $\gamma$.

**for** $t = 1 \ldots E; i = 1 \ldots M$ **do**

  $\ell_{NT} = \ell(x, y; \theta)$;

  **if** $\text{DOM}_{\text{RE}}$ **and** $t > \mathcal{K}$ **then**

    **if** *Natural Training* **then**

      $\theta = \theta - \nabla_\theta \left( \ell_{NT}(\ell_{NT} > \mathcal{T}) \right)$;

    **else if** *Adversarial Training* **then**

      $\ell_{AT} = \ell(x + \delta, y; \theta)$;

      $\theta = \theta - \nabla_\theta \left( \ell_{AT}(\ell_{NT} > \mathcal{T}) \right)$;

  **else if** $\text{DOM}_{\text{DA}}$ **and** $t > \mathcal{K}$ **then**

    **while** $n <= \gamma$ **do**

      **if** $\ell(\mathcal{DA}(x(\ell_{NT} < \mathcal{T})), y; \theta) > \mathcal{T}$ **then**

        $x_{DA}(\ell_{NT} < \mathcal{T}) = \mathcal{DA}(x(\ell_{NT} < \mathcal{T}))$ **and break**;

      **else**

        $x_{DA}(\ell_{NT} < \mathcal{T}) = x(\ell_{NT} < \mathcal{T}) * (1 - \beta) + \mathcal{DA}(x(\ell_{NT} < \mathcal{T})) * \beta$;

    **if** *Natural Training* **then**

      $\ell_{DA-NT} = \ell(x_{DA}, y; \theta)$;

      $\theta = \theta - \nabla_\theta (\ell_{DA-NT})$;

    **else if** *Adversarial Training* **then**

      $\ell_{DA-AT} = \ell(x_{DA} + \delta_{DA}, y; \theta)$;

      $\theta = \theta - \nabla_\theta (\ell_{DA-AT})$;

  **else**

    # Standard optimize network parameter $\theta$ according to training paradigm.

---

data augmentation denoted as $\text{DOM}_{\text{RE}}$ and $\text{DOM}_{\text{DA}}$, respectively. For $\text{DOM}_{\text{RE}}$, we adopt a straightforward approach to remove all high-confidence patterns without distinguishing over-memorization and normal-memorization. This depends on the observation that DNNs exhibit a significantly persistent memory for over-memorization patterns, as evidenced in Figure 2. As training progresses, we expect the loss of normal-memorization patterns to gradually increase, eventually surpassing the threshold and prompting the model to relearn. In contrast, the loss for over-memorization patterns is unlikely to notably increase with further training, hindering their likelihood of being relearned.

On the other hand, $\text{DOM}_{\text{DA}}$ utilizes data augmentation techniques to weaken the model's confidence in over-memorization patterns. Nonetheless, research by Rice et al. (2020); Zhang et al. (2022) have shown that the ability of original data augmentation is limited for mitigating RO and CO. From the perspective of over-memorization, we employ iterative data augmentation on high-confidence patterns to maximization reduce the model's reliance on them, thereby effectively mitigating overfitting. The implementation of the proposed framework DOM is summarized in Algorithm 1.

## 4 EXPERIMENTS

In this section, we conduct extensive experiments to verify the effectiveness of DOM, including experiment settings (Section 4.1), performance evaluation (Section 4.2), and ablation studies (Section 4.3).

### 4.1 EXPERIMENT SETTINGS

**Data Argumentation.** The standard data augmentation techniques random cropping and horizontal flipping are applied in all configurations. For $\text{DOM}_{\text{DA}}$, we use two popular techniques, AUG-MIX (Hendrycks et al., 2019) and RandAugment (Cubuk et al., 2020).

**Adversarial Paradigm.** We follow the widely-used configurations, setting the perturbation budget as $\epsilon = 8/255$ and adopting the threat model as $L_\infty$. For adversarial training, we employ the default PGD-10 (Madry et al., 2018) and RS-FGSM (Wong et al., 2019) to generate the multi-step and single-step adversarial perturbation, respectively. For the adversarial test, we use the PGD-20 and Auto Attack (Croce & Hein, 2020) to evaluate model robustness.

**Datasets and Model Architectures.** We conducted extensive experiments on the benchmark datasets Cifar-10/100 (Krizhevsky et al., 2009), SVHN (Netzer et al., 2011) and Tiny-ImageNet (Netzer et al.,

Table 1. The CIFAR-10/100 hyperparameter settings are divided by slashes. The 1st to 3rd columns are general settings, and the 4th to 9th columns are DOM settings.

| Method | Learning rate (l.r. decay) | Training Epoch | Warm-up Epoch | Loss Threshold | AUGMIX Strength | AUGMIX Iteration | RandAugment Strength | RandAugment Iteration |
|---|---|---|---|---|---|---|---|---|
| Natural | 0.1 (150, 225) | 300 | 150 | 0.2/0.45 | 50% | 3/2 | 10% | 3/2 |
| PGD-10 | 0.1 (100, 150) | 200 | 100 | 1.5/4.0 | 50% | 2 | 0% | 2 |
| RS-FGSM | 0.0-0.2 (cyclical) | 100/50 | 50/25 | 2.0/4.6 | 50% | 5 | 10% | 3 |

Table 2. Natural training test error on CIFAR10/100. The results are averaged over 3 random seeds and reported with the standard deviation.

| Network | Method | CIFAR10 | | | CIFAR100 | | |
|---|---|---|---|---|---|---|---|
| | | Best ($\downarrow$) | Last ($\downarrow$) | Diff ($\downarrow$) | Best ($\downarrow$) | Last ($\downarrow$) | Diff ($\downarrow$) |
| PreactResNet-18 | Baseline | $4.70 \pm 0.09$ | $4.84 \pm 0.04$ | -0.14 | $\mathbf{21.32 \pm 0.03}$ | $21.61 \pm 0.03$ | -0.29 |
| | + DOM$_\mathrm{RE}$ | $\mathbf{4.55 \pm 0.19}$ | $\mathbf{4.63 \pm 0.19}$ | $\mathbf{-0.08}$ | $21.35 \pm 0.20$ | $\mathbf{21.44 \pm 0.06}$ | $\mathbf{-0.09}$ |
| | + AUGMIX | $4.35 \pm 0.18$ | $4.52 \pm 0.01$ | -0.17 | $21.79 \pm 0.32$ | $22.06 \pm 0.35$ | -0.27 |
| | + DOM$_\mathrm{DA}$ | $\mathbf{4.13 \pm 0.14}$ | $\mathbf{4.24 \pm 0.02}$ | $\mathbf{-0.11}$ | $\mathbf{21.67 \pm 0.06}$ | $\mathbf{21.79 \pm 0.30}$ | $\mathbf{-0.12}$ |
| | + RandAugment | $4.02 \pm 0.08$ | $4.31 \pm 0.06$ | -0.29 | $21.13 \pm 0.05$ | $21.61 \pm 0.11$ | -0.48 |
| | + DOM$_\mathrm{DA}$ | $\mathbf{3.96 \pm 0.08}$ | $\mathbf{4.07 \pm 0.13}$ | $\mathbf{-0.11}$ | $\mathbf{21.11 \pm 0.09}$ | $\mathbf{21.49 \pm 0.06}$ | $\mathbf{-0.38}$ |
| WideResNet-34 | Baseline | $3.71 \pm 0.12$ | $3.86 \pm 0.19$ | -0.15 | $\mathbf{18.24 \pm 0.19}$ | $18.57 \pm 0.06$ | -0.33 |
| | + DOM$_\mathrm{RE}$ | $\mathbf{3.63 \pm 0.13}$ | $\mathbf{3.75 \pm 0.11}$ | $\mathbf{-0.12}$ | $18.30 \pm 0.04$ | $\mathbf{18.52 \pm 0.07}$ | $\mathbf{-0.22}$ |
| | + AUGMIX | $3.43 \pm 0.05$ | $3.69 \pm 0.13$ | -0.26 | $18.23 \pm 0.18$ | $18.43 \pm 0.21$ | -0.20 |
| | + DOM$_\mathrm{DA}$ | $\mathbf{3.42 \pm 0.19}$ | $\mathbf{3.58 \pm 0.03}$ | $\mathbf{-0.16}$ | $\mathbf{18.18 \pm 0.18}$ | $\mathbf{18.36 \pm 0.01}$ | $\mathbf{-0.18}$ |
| | + RandAugment | $3.20 \pm 0.08$ | $3.44 \pm 0.08$ | -0.24 | $\mathbf{17.61 \pm 0.10}$ | $17.97 \pm 0.02$ | -0.36 |
| | + DOM$_\mathrm{DA}$ | $\mathbf{2.98 \pm 0.02}$ | $\mathbf{3.20 \pm 0.12}$ | $\mathbf{-0.22}$ | $17.88 \pm 0.18$ | $\mathbf{17.93 \pm 0.01}$ | $\mathbf{-0.05}$ |

2011). The settings and results for SVHN and Tiny-ImageNet are provided in Appendix C and Appendix D, respectively. We train the PreactResNet-18 (He et al., 2016), WideResNet-34 (Zagoruyko & Komodakis, 2016) and ViT-small (Dosovitskiy et al., 2020) architectures on these datasets by utilizing the SGD optimizer with a momentum of 0.9 and weight decay of $5 \times 10^{-4}$. The results of ViT-small can be found in Appendix E. Other hyperparameters setting, including learning rate schedule, training epochs E, warm-up epoch $\mathcal{K}$, loss threshold $\mathcal{T}$, data augmentation strength $\beta$ and data augmentation iteration $\gamma$ are summarized in Table 1. We also evaluate our methods on the gradual learning rate schedule, as shown in Appendix F.

## 4.2 PERFORMANCE EVALUATION

**Natural Training Results.** In Table 2, we present an evaluation of the proposed framework against competing baselines on CIFAR-10/100 datasets. We report the test accuracy at both the highest (Best) and final (Last) checkpoint during training, as well as the generalization gap between them (Diff). Firstly, we can observe that $\mathrm{DOM}_\mathrm{RE}$, which is trained on a strict subset of natural patterns, can consistently outperform baselines at the final checkpoint. Secondly, $\mathrm{DOM}_\mathrm{DA}$ can achieve superior performance at the both highest and final checkpoints. It's worth noting that $\mathrm{DOM}_\mathrm{DA}$ applies data augmentation to limited epochs and training patterns. Finally and most importantly, both $\mathrm{DOM}_\mathrm{RE}$ and $\mathrm{DOM}_\mathrm{DA}$ can successfully reduce the model's generalization gap, which substantiates our perspective that over-memorization hinders model generalization, and preventing it can alleviate overfitting.

**Adversarial Training Results.** To further explore the over-memorization, we extend our framework to both multi-step and single-step AT. Importantly, the detection of over-memorization adversarial patterns relies exclusively on the loss of the corresponding natural pattern. From Table 3, it's evident that both $\mathrm{DOM}_\mathrm{RE}$ and $\mathrm{DOM}_\mathrm{DA}$ are effective in eliminating RO under PGD-20 attack. However, under Auto Attack, the $\mathrm{DOM}_\mathrm{DA}$ remains its superior robustness, whereas $\mathrm{DOM}_\mathrm{RE}$ is comparatively weaker. This difference in Auto Attack could be attributed to $\mathrm{DOM}_\mathrm{RE}$ directly removing training patterns, potentially ignoring some useful information. Table 4 illustrates that both $\mathrm{DOM}_\mathrm{RE}$ and $\mathrm{DOM}_\mathrm{DA}$ are effective in mitigating CO. However, the proposed framework shows its limitation in preventing CO when using $\mathrm{DOM}_\mathrm{DA}$ with AUGMIX on CIFAR100. This result could stem from the weakness of the original data augmentation method, which remains inability to break over-memorization even after the framework's iterative operation.

Table 3. Multi-step adversarial training test accuracy on CIFAR10/100. The results are averaged over 3 random seeds and reported with the standard deviation.

| Dataset | Method | Best | | | Last | | |
|---|---|---|---|---|---|---|---|
| | | Natural (↑) | PGD-20 (↑) | Auto Attack (↑) | Natural (↑) | PGD-20 (↑) | Auto Attack (↑) |
| CIFAR10 | Baseline | **81.70 ± 0.48** | 52.33 ± 0.25 | **48.02 ± 0.49** | **83.59 ± 0.15** | 45.16 ± 1.20 | **42.70 ± 1.16** |
| | + $\text{DOM}_{\text{RE}}$ | 80.23 ± 0.06 | **55.48 ± 0.37** | 42.87 ± 0.32 | 80.66 ± 0.33 | **52.52 ± 1.29** | 32.90 ± 1.02 |
| | + AUGMIX | 79.92 ± 0.77 | 52.76 ± 0.07 | 47.91 ± 0.21 | 84.07 ± 0.39 | 47.71 ± 1.50 | 44.71 ± 1.06 |
| | + $\text{DOM}_{\text{DA}}$ | **80.87 ± 0.98** | **53.54 ± 0.15** | **47.98 ± 0.14** | **84.15 ± 0.26** | **49.31 ± 0.83** | **45.51 ± 0.85** |
| | + RandAugment | 82.73 ± 0.38 | 52.73 ± 0.20 | 48.39 ± 0.03 | 82.40 ± 1.46 | 47.84 ± 1.69 | 44.27 ± 1.63 |
| | + $\text{DOM}_{\text{DA}}$ | **83.49 ± 0.69** | **52.83 ± 0.06** | **48.41 ± 0.28** | **83.74 ± 0.57** | **50.39 ± 0.91** | **46.62 ± 0.69** |
| CIFAR100 | Baseline | **56.04 ± 0.33** | 29.32 ± 0.04 | **25.19 ± 0.23** | **57.09 ± 0.32** | 21.92 ± 0.53 | **19.81 ± 0.49** |
| | + $\text{DOM}_{\text{RE}}$ | 52.70 ± 0.71 | **29.45 ± 0.33** | 20.41 ± 0.56 | 52.67 ± 0.96 | **25.14 ± 0.39** | 17.59 ± 0.35 |
| | + AUGMIX | 52.46 ± 0.73 | 29.54 ± 0.24 | 24.15 ± 0.14 | 57.53 ± 0.62 | 24.15 ± 0.10 | 21.22 ± 0.08 |
| | + $\text{DOM}_{\text{DA}}$ | **56.07 ± 0.23** | **29.81 ± 0.07** | **25.09 ± 0.02** | **57.70 ± 0.02** | **24.80 ± 0.36** | **21.84 ± 0.30** |
| | + RandAugment | 55.12 ± 1.33 | 28.62 ± 0.04 | 23.80 ± 0.21 | 55.71 ± 1.62 | 23.10 ± 1.32 | 20.03 ± 1.00 |
| | + $\text{DOM}_{\text{DA}}$ | **55.20 ± 1.34** | **30.01 ± 0.57** | **24.10 ± 0.88** | **56.21 ± 1.93** | **25.84 ± 0.39** | **20.79 ± 0.96** |

Table 4. Single-step adversarial training final checkpoint's test accuracy on CIFAR10/100. The results are averaged over 3 random seeds and reported with the standard deviation.

| Method | CIFAR10 | | | CIFAR100 | | |
|---|---|---|---|---|---|---|
| | Natural (↑) | PGD-20 (↑) | Auto Attack (↑) | Natural (↑) | PGD-20 (↑) | Auto Attack (↑) |
| Baseline | **87.77 ± 3.02** | 0.00 ± 0.00 | 0.00 ± 0.00 | **60.28 ± 3.34** | 0.00 ± 0.00 | 0.00 ± 0.00 |
| + $\text{DOM}_{\text{RE}}$ | 71.66 ± 0.29 | **47.09 ± 0.36** | **17.10 ± 0.82** | 26.39 ± 1.06 | **12.68 ± 0.62** | **7.65 ± 0.59** |
| + AUGMIX | **88.82 ± 0.99** | 0.00 ± 0.00 | 0.00 ± 0.00 | 48.05 ± 4.84 | 0.00 ± 0.00 | 0.00 ± 0.00 |
| + $\text{DOM}_{\text{DA}}$ | 84.31 ± 0.59 | **45.15 ± 0.06** | **41.16 ± 0.11** | **63.03 ± 0.19** | 0.00 ± 0.00 | 0.00 ± 0.00 |
| + RandAugment | **84.63 ± 0.83** | 0.00 ± 0.00 | 0.00 ± 0.00 | **59.39 ± 0.62** | 0.00 ± 0.00 | 0.00 ± 0.00 |
| + $\text{DOM}_{\text{DA}}$ | 84.52 ± 0.28 | **50.10 ± 1.53** | **42.53 ± 1.41** | 55.09 ± 1.73 | **27.44 ± 0.12** | **21.38 ± 0.78** |

**Overall Results.** In summary, the DOM framework can effectively mitigate different types of over-fitting by consistently preventing the shared behaviour over-memorization, which first-time employs a unified perspective to understand and address overfitting across different training paradigms.

## 4.3 ABLATION STUDIES

In this section, we investigate the impacts of algorithmic components using PreactResNet-18 on CIFAR10. For the loss threshold and warm-up epoch selection, we employ $\text{DOM}_{\text{RE}}$, while for data augmentation strength and iteration selection, we use $\text{DOM}_{\text{DA}}$ with AUGMIX in the context of NT. When tuning a specific hyperparameter, we keep other hyperparameters fixed.

**Loss Threshold Selection.** To investigate the role of loss threshold, we present the variations in test error across three training paradigms. As depicted in Figure 5 (a: left), we can observe that employing a small threshold might not effectively filter out over-memorization patterns, resulting in suboptimal generalization performance. On the other hand, adopting a larger threshold might lead to the exclusion lot of training patterns, consequently resulting in the model underfitting. In light of this trade-off, we set the loss threshold as 0.2 for NT. Interestingly, this trade-off does not seem to exist in the context of AT, where higher loss thresholds tend to result in higher PGD robustness, as shown in Figure 5 (a: middle and right). Nevertheless, the above experiments indicate that this approach could also increase the vulnerability to Auto Attack. Hence, determining an appropriate loss threshold is critical for all training paradigms. We also evaluate our methods on unified adaptive loss threshold (Berthelot et al., 2021; Li et al., 2023) as shown in Appendix G.

**Warm-Up Epoch Selection.** The observations from Figure 5 (b: left) indicate that a short warm-up period might hinder the DNNs from learning essential information, leading to a decline in the performance. Conversely, a longer warm-up period cannot promptly prevent the model from over-memorizing training patterns, which also results in compromised generalization performance. Based on this observation, we simply align the warm-up epoch with the model's first learning rate decay.

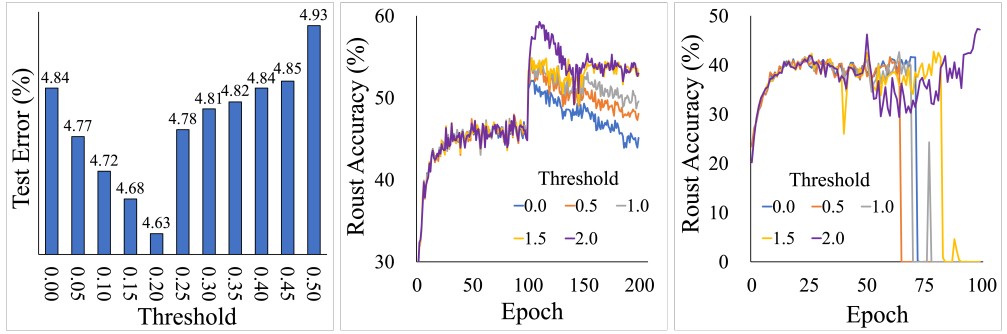

(a) The role of loss threshold in natural, multi-step and single-step adversarial training(from left to right).

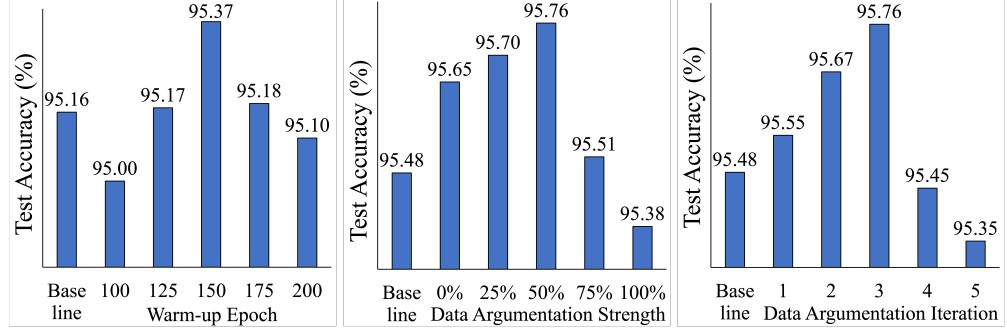

(b) The role of warm-up epoch, data argumentation strength and iteration (from left to right).

Figure 5. Ablation Study

**Data Augmentation Strength and Iteration Selection.** We also examine the impact of data augmentation strengths and iterations, as shown in Figure 5 (b: middle and right). We can observe that, even when the augmentation strength is set to 0% or the number of iterations is limited to 1, our approach can still outperform the baseline (AUGMIX). Moreover, both insufficient (weak strengths or few iterations) and aggressive (strong strengths or excessive iterations) augmentations will lead to subpar performance. This is due to insufficient augmentations limiting the pattern transformation to diverse styles, while aggressive augmentations could exacerbate classification difficulty and even distort the semantic information (Bai et al., 2022; Huang et al., 2023b). Therefore, we select the augmentation strength as 50% and iteration as 3 to achieve the optimal performance. The computational overhead analysis can be found in the Appendix H.

## 5 CONCLUSION

Previous research has made significant progress in understanding and addressing natural, robust, and catastrophic overfitting, individually. However, the common understanding and solution for these overfitting have remained unexplored. To the best of our knowledge, our study first-time bridges this gap by providing a unified perspective on overfitting. Specifically, we examine the memorization effect in deep neural networks, and identify a shared behaviour termed over-memorization across various training paradigms. This behaviour is characterized by the model suddenly becoming high-confidence predictions and retaining a persistent memory in certain training patterns, subsequently resulting in a decline in generalization ability. Our findings also reveal that when the model over-memorizes an adversarial pattern, it tends to simultaneously exhibit high-confidence in predicting the corresponding natural pattern. Building on the above insights, we propose a general framework named *Distraction Over-Memorization* (DOM), designed to holistically mitigate different types of overfitting by proactively preventing over-memorization training patterns.

**Limitations.** This paper offers a shared comprehension and remedy for overfitting across various training paradigms. Nevertheless, a detailed theoretical analysis of the underlying mechanisms among these overfitting types remains an open question for future research. Besides, the effectiveness of the proposed $\text{DOM}_{\text{DA}}$ method is dependent on the quality of the original data augmentation technique, which could potentially limit its applicability in some scenarios.

ACKNOWLEDGMENTS

The authors would like to thank Huaxi Huang, reviewers and area chair for their helpful and valuable comments. Bo Han was supported by the NSFC General Program No. 62376235, Guangdong Basic and Applied Basic Research Foundation Nos. 2022A1515011652 and 2024A1515012399, HKBU Faculty Niche Research Areas No. RC-FNRA-IG/22-23/SCI/04, and HKBU CSD Departmental Incentive Scheme. Tongliang Liu is partially supported by the following Australian Research Council projects: FT220100318, DP220102121, LP220100527, LP220200949, and IC190100031.

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

## A    DETAILED EXPERIMENT SETTINGS

In Section 3, we conducted all experiments on the CIFAR-10 dataset using PreactResNet-18. We analyzed the proportion of natural and adversarial patterns by examining the respective natural and adversarial training loss. In Section 3.1, we categorized between original and transformed high-confidence patterns using an auxiliary model, which was saved at the first learning rate decay (150th epoch). In Section 3.2 Figure 4, we grouped adversarial patterns based on their corresponding natural training loss, employing a loss threshold of 1.5.

## B    TRADES RESULTS

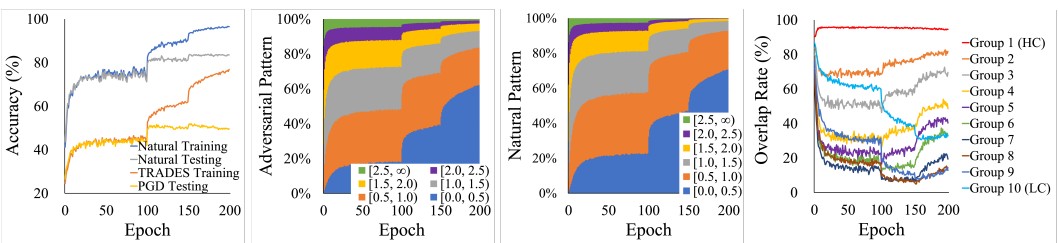

Figure 6. TRADES adversarial training. 1st Panel: The training and test accuracy of adversarial training. 2nd/3rd Panel: Proportion of adversarial/natural patterns based on varying training loss ranges. 4th Panel: The overlap rate between natural and adversarial patterns grouped by training loss rankings.

We further explored this observation in the TRADES-trained model, which encounters natural patterns during the training process. From Figure 6, we can observe that TRADES demonstrates a consistent memory tendency with PGD in the over-memorization samples. This tendency manifests as, when DNNs over-memorize certain adversarial patterns, they tend to simultaneously exhibit high-confidence in predicting the corresponding natural patterns.

## C    SETTINGS AND RESULTS ON SVHN

**SVHN Settings.** In accordance with the settings of Rice et al. (2020); Wong et al. (2019), we adopt a gradually increasing perturbation step size in the initial 10 and 5 epochs for multi-step and single-step AT, respectively. In the meantime, the PGD step size is set as $\alpha = 1/255$. Other hyperparameters setting, including learning rate schedule, training epochs E, loss threshold $\mathcal{T}$, warm-up epoch $\mathcal{K}$, data augmentation strength $\beta$ and data augmentation times $\gamma$ are summarized in Table 5.

Table 5. The SVHN hyperparameter settings. The 1st to 3rd columns are general settings, and the 4th to 9th columns are DOM settings.

| Method | l.r. (l.r. decay) | Training epoch | Warm-up epoch | Loss threshold | AUGMIX strength | AUGMIX times | RandAugment strength | RandAugment times |
|---|---|---|---|---|---|---|---|---|
| Natural | 0.01 (150, 225) | 300 | 150 | 0.02 | 50% | 3 | 50% | 3 |
| PGD-10 | 0.01 (100, 150) | 200 | 100 | 0.75 | 50% | 4 | 25% | 2 |
| RS-FGSM | 0.0-0.01 (cyclical) | 20 | 10 | 1.0 | 50% | 4 | 10% | 4 |

**SVHN Results.** To verify the pervasive applicability of our perspective and method, we extend the DOM framework to the SVHN dataset. The results for NT, multi-step AT, and single-step AT are reported in Table 6, Table 7, and Table 8, respectively. From Table 6, it is clear that both $\mathrm{DOM_{RE}}$ and $\mathrm{DOM_{DA}}$ not only achieve superior performance at both the highest and final checkpoints but also succeed in reducing the generalization gap, thereby effectively mitigating NO.

Table 6. Natural training test error on SVHN. The results are averaged over 3 random seeds and reported with the standard deviation.

| Method | PreactResNet-18 | | | WideResNet-34 | | |
|---|---|---|---|---|---|---|
| | Best ($\downarrow$) | Last ($\downarrow$) | Diff ($\downarrow$) | Best ($\downarrow$) | Last ($\downarrow$) | Diff ($\downarrow$) |
| Baseline | $3.45 \pm 0.09$ | $3.53 \pm 0.10$ | -0.08 | $3.36 \pm 0.53$ | $3.61 \pm 0.07$ | -0.25 |
| + DOM$_{RE}$ | $\mathbf{3.43 \pm 0.02}$ | $\mathbf{3.50 \pm 0.05}$ | **-0.07** | $\mathbf{2.75 \pm 0.01}$ | $\mathbf{2.87 \pm 0.05}$ | **-0.12** |
| + AUGMIX | $\mathbf{3.44 \pm 0.02}$ | $3.52 \pm 0.05$ | -0.08 | $\mathbf{3.06 \pm 0.08}$ | $3.17 \pm 0.08$ | -0.11 |
| + DOM$_{DA}$ | $3.44 \pm 0.06$ | $\mathbf{3.50 \pm 0.05}$ | **-0.06** | $3.10 \pm 0.02$ | $\mathbf{3.15 \pm 0.04}$ | **-0.05** |
| + RandAugment | $3.00 \pm 0.07$ | $3.12 \pm 0.02$ | -0.12 | $2.65 \pm 0.01$ | $2.84 \pm 0.09$ | -0.19 |
| + DOM$_{DA}$ | $\mathbf{2.98 \pm 0.01}$ | $\mathbf{3.07 \pm 0.03}$ | **-0.09** | $\mathbf{2.60 \pm 0.08}$ | $\mathbf{2.77 \pm 0.10}$ | **-0.17** |

From Table 7, we can observe that the $\text{DOM}_{RE}$ can demonstrate improved robustness against PGD, while $\text{DOM}_{DA}$ shows better robustness against both PGD and Auto Attack at the final checkpoint, which confirms their effectiveness in eliminating RO.

Table 7. Multi-step adversarial training test accuracy on SVHN. The results are averaged over 3 random seeds and reported with the standard deviation.

| Method | Best | | | Last | | |
|---|---|---|---|---|---|---|
| | Natural ($\uparrow$) | PGD-20 ($\uparrow$) | Auto Attack ($\uparrow$) | Natural ($\uparrow$) | PGD-20 ($\uparrow$) | Auto Attack ($\uparrow$) |
| Baseline | $91.00 \pm 0.41$ | $53.50 \pm 0.35$ | $\mathbf{45.45 \pm 0.23}$ | $\mathbf{92.83 \pm 0.15}$ | $48.32 \pm 0.24$ | $\mathbf{38.51 \pm 0.43}$ |
| + DOM$_{RE}$ | $\mathbf{91.40 \pm 0.51}$ | $\mathbf{54.31 \pm 0.62}$ | $41.74 \pm 0.55$ | $91.53 \pm 0.41$ | $\mathbf{49.59 \pm 1.37}$ | $31.22 \pm 1.00$ |
| + AUGMIX | $92.44 \pm 1.02$ | $53.75 \pm 0.53$ | $\mathbf{46.29 \pm 0.83}$ | $\mathbf{93.79 \pm 0.35}$ | $51.12 \pm 0.29$ | $42.73 \pm 0.74$ |
| + DOM$_{DA}$ | $\mathbf{92.73 \pm 0.51}$ | $\mathbf{55.31 \pm 0.23}$ | $45.72 \pm 1.07$ | $92.05 \pm 0.89$ | $\mathbf{53.64 \pm 0.42}$ | $\mathbf{43.14 \pm 0.65}$ |
| + RandAugment | $93.01 \pm 0.11$ | $54.06 \pm 0.25$ | $\mathbf{46.02 \pm 0.06}$ | $\mathbf{93.38 \pm 0.81}$ | $52.36 \pm 0.36$ | $44.28 \pm 1.21$ |
| + DOM$_{DA}$ | $\mathbf{93.16 \pm 0.80}$ | $\mathbf{56.13 \pm 0.30}$ | $44.82 \pm 1.27$ | $92.81 \pm 1.31$ | $\mathbf{54.01 \pm 1.52}$ | $\mathbf{44.59 \pm 0.67}$ |

Table 8 indicates that both $\text{DOM}_{RE}$ and $\text{DOM}_{DA}$ can effectively mitigate CO in all test scenarios. Overall, the above results not only emphasize the extensiveness of over-memorization, but also highlight the effectiveness of the DOM across diverse datasets.

Table 8. Single-step adversarial training final checkpoint's test accuracy on SVHN. The results are averaged over 3 random seeds and reported with the standard deviation.

| Method | Natural ($\uparrow$) | PGD-20 ($\uparrow$) | Auto Attack ($\uparrow$) |
|---|---|---|---|
| Baseline | $\mathbf{98.21 \pm 0.35}$ | $0.02 \pm 0.03$ | $0.00 \pm 0.00$ |
| + DOM$_{RE}$ | $89.96 \pm 0.55$ | $\mathbf{47.92 \pm 0.63}$ | $\mathbf{32.22 \pm 1.00}$ |
| + AUGMIX | $\mathbf{98.09 \pm 0.23}$ | $0.07 \pm 0.03$ | $0.00 \pm 0.01$ |
| + DOM$_{DA}$ | $90.67 \pm 0.49$ | $\mathbf{49.88 \pm 0.37}$ | $\mathbf{37.67 \pm 0.12}$ |
| + RandAugment | $\mathbf{98.04 \pm 0.60}$ | $0.35 \pm 0.22$ | $0.05 \pm 0.04$ |
| + DOM$_{DA}$ | $85.88 \pm 3.02$ | $\mathbf{51.57 \pm 1.79}$ | $\mathbf{36.69 \pm 1.55}$ |

# D   SETTINGS AND RESULTS ON TINY-IMAGENET

We also verified the effectiveness of our method on the larger-scale dataset Tiny-ImageNet (Netzer et al., 2011). We set the loss threshold $\mathcal{T}$ to 0.2, and other hyperparameters remain as the original settings.

Table 9. Tiny-ImageNet: The natural training test error at the best and last checkpoint using PreactResNet-18. The results are averaged over 3 random seeds and reported with the standard deviation.

| Method | Best ($\downarrow$) | Last ($\downarrow$) | Diff ($\downarrow$) |
|---|---|---|---|
| Baseline | 35.03 ± 0.03 | 35.24 ± 0.02 | -0.21 |
| + $\text{DOM}_{\text{RE}}$ | **34.89 ± 0.02** | **34.99 ± 0.01** | **-0.10** |
| + AUGMIX | 34.98 ± 0.02 | 35.15 ± 0.03 | -0.17 |
| + $\text{DOM}_{\text{DA}}$ | **34.56 ± 0.04** | **34.77 ± 0.02** | **-0.21** |
| + RandAugment | 33.46 ± 0.05 | 33.89 ± 0.06 | -0.45 |
| + $\text{DOM}_{\text{DA}}$ | **33.44 ± 0.04** | **33.59 ± 0.02** | **-0.20** |

Table 9 illustrates the effectiveness of our method, $DOM_{RE}$ and $DOM_{DA}$, on the Tiny-ImageNet dataset. These results indicate that preventing over-memorization can improve model performance and reduce the generalization gap on large-scale datasets.

## E  SETTINGS AND RESULTS ON VIT

We have validated the effectiveness of our method within CNN-based architectures, demonstrating its ability to alleviate overfitting by preventing over-memorization. To further substantiate our perspective, we verify our method on the Transformer-based architecture. Constrained by computational resources, we trained a ViT-small model (Dosovitskiy et al., 2020), initializing it with pre-trained weights from the Timm Python library. The training spanned 100 epochs, starting with an initial learning rate of 0.001 and divided by 10 at the 50th and 75th epochs. We set the batch size to 64 and the loss threshold $\mathcal{T}$ to 0.1, maintaining other hyperparameters as the original settings.

Table 10. Vit: The natural training test error at the best and last checkpoint on CIFAR 10. The results are averaged over 3 random seeds and reported with the standard deviation.

| Method | Best ($\downarrow$) | Last ($\downarrow$) | Diff ($\downarrow$) |
|---|---|---|---|
| Baseline | 1.49 ± 0.01 | 1.75 ± 0.01 | -0.26 |
| + $\text{DOM}_{\text{RE}}$ | **1.47 ± 0.02** | **1.67 ± 0.01** | **-0.20** |
| + AUGMIX | 1.24 ± 0.01 | 1.29 ± 0.01 | **-0.05** |
| + $\text{DOM}_{\text{DA}}$ | **1.20 ± 0.01** | **1.27 ± 0.01** | -0.07 |
| + RandAugment | 1.21 ± 0.01 | 1.27 ± 0.02 | -0.06 |
| + $\text{DOM}_{\text{DA}}$ | **1.17 ± 0.01** | **1.22 ± 0.01** | **-0.05** |

Table 10 shows the effectiveness of our method on the Transformer-based architecture. By mitigating over-memorization, both $DOM_{RE}$ and $DOM_{DA}$ not only improve model performance at both the best and last checkpoints, but also contribute to alleviating overfitting.

## F  GRADUALLY LEARNING RATE RESULTS

To further assess our method, we conducted experiments using a gradual learning rate schedule (Smith, 2017) in natural training. We set the cyclical learning rate schedule with 300 epochs, reaching the maximum learning rate of 0.2 at the midpoint of 150 epochs.

From Table 11, it is apparent that although the cyclical learning rate reduces the model's generalization gap, it also leads to a reduction in performance compared to the step learning rate. Nevertheless, our method consistently showcases its effectiveness in improving model performance and completely eliminating the generalization gap by mitigating over-memorization.

Table 11. Cyclical learning rate: The natural training test error at the best and last checkpoint on CIFAR 10 using PreactResNet-18. The results are averaged over 3 random seeds and reported with the standard deviation.

| Method | Best ($\downarrow$) | Last ($\downarrow$) | Diff ($\downarrow$) |
|---|---|---|---|
| Baseline | $4.80 \pm 0.03$ | $4.89 \pm 0.03$ | -0.09 |
| + $\text{DOM}_{RE}$ | $\mathbf{4.79 \pm 0.02}$ | $\mathbf{4.79 \pm 0.02}$ | **-0.00** |
| + AUGMIX | $4.75 \pm 0.01$ | $4.79 \pm 0.02$ | -0.02 |
| + $\text{DOM}_{DA}$ | $\mathbf{4.49 \pm 0.01}$ | $\mathbf{4.49 \pm 0.01}$ | **-0.00** |
| + RandAugment | $4.41 \pm 0.03$ | $4.42 \pm 0.01$ | -0.01 |
| + $\text{DOM}_{DA}$ | $\mathbf{4.25 \pm 0.01}$ | $\mathbf{4.25 \pm 0.01}$ | **-0.00** |

## G  ADAPTIVE LOSS THRESHOLD

By utilizing the fixed loss threshold DOM, we have effectively verified and mitigated over-memorization, which negatively impacts DNNs' generalization ability. However, as a general framework, finding an optimal loss threshold for different paradigms and datasets can be cumbersome. To address this challenge, we propose to use a general and unified loss threshold applicable across all experimental settings. Specifically, we utilize an adaptive loss threshold (Berthelot et al., 2021), whose value is dependent on the loss of the model's current training batch. For all experiments, we set this adaptive loss threshold $\mathcal{T}$ to 40%, maintaining other hyperparameters as the original settings.

Table 12. Adaptive loss threshold: The natural and PGD-20 test error for natural training (NT) and adversarial training (AT) using PreactResNet-18. The results are averaged over 3 random seeds and reported with the standard deviation.

| Dataset | Paradigm | Method | Best ($\downarrow$) | Last ($\downarrow$) | Diff ($\downarrow$) |
|---|---|---|---|---|---|
| | | Baseline | $4.70 \pm 0.09$ | $4.84 \pm 0.04$ | -0.14 |
| | | + $\text{DOM}_{RE}$ | $\mathbf{4.62 \pm 0.06}$ | $\mathbf{4.68 \pm 0.02}$ | **-0.06** |
| CIFAR10 | NT | + AUGMIX | $4.35 \pm 0.18$ | $4.52 \pm 0.01$ | -0.17 |
| | | + $\text{DOM}_{DA}$ | $\mathbf{4.24 \pm 0.10}$ | $\mathbf{4.37 \pm 0.08}$ | **-0.13** |
| | | + RandAugment | $4.02 \pm 0.08$ | $4.31 \pm 0.06$ | -0.29 |
| | | + $\text{DOM}_{DA}$ | $\mathbf{3.86 \pm 0.05}$ | $\mathbf{3.94 \pm 0.06}$ | **-0.08** |
| CIFAR100 | NT | Baseline | $21.32 \pm 0.03$ | $21.59 \pm 0.03$ | -0.27 |
| | | + $\text{DOM}_{RE}$ | $\mathbf{21.20 \pm 0.07}$ | $\mathbf{21.43 \pm 0.04}$ | **-0.23** |
| CIFAR10 | Multi-step AT | Baseline | $47.67 \pm 0.25$ | $54.84 \pm 1.20$ | -7.17 |
| | | + $\text{DOM}_{RE}$ | $\mathbf{46.57 \pm 0.64}$ | $\mathbf{52.83 \pm 0.28}$ | **-6.26** |
| CIFAR10 | Single-step AT | Baseline | $57.83 \pm 1.24$ | $100.00 \pm 0.00$ | -42.17 |
| | | + $\text{DOM}_{RE}$ | $\mathbf{54.52 \pm 0.57}$ | $\mathbf{56.36 \pm 0.92}$ | **-1.84** |

Table 12 demonstrates the effectiveness of the adaptive loss threshold across different paradigms and datasets. This threshold can not only consistently identify over-memorization patterns and mitigate overfitting, but also be easily transferable without the need for hyperparameter tuning.

## H  COMPUTATIONAL OVERHEAD

We analyze the extra computational overhead incurred by the DOM framework. Notably, both $DOM_{RE}$ and $DOM_{DA}$ are implemented after the warm-up period (half of the training epoch).

Based on Table 13, we can observe that $DOM_{RE}$ does not involve any additional computational overhead. Although $DOM_{DA}$ require iterative forward propagation, its overall training time does not

Table 13. The training cost (epoch/second) on CIFAR10 using PreactResNet-18 with a single NVIDIA RTX 4090 GPU.

| Method | Before warm-up ($\downarrow$) | After warm-up ($\downarrow$) | Overall ($\downarrow$) |
|---|---|---|---|
| Baseline | 6.28 | 6.26 | 6.27 |
| + $\text{DOM}_{\text{RE}}$ | 6.28 | 6.28 | 6.28 |
| + AUGMIX | 12.55 | 12.76 | 12.66 |
| + $\text{DOM}_{\text{DA}}$ | 6.28 | 28.45 | 17.37 |
| + RandAugment | 8.29 | 8.27 | 8.28 |
| + $\text{DOM}_{\text{DA}}$ | 6.24 | 12.75 | 9.50 |

significantly increase, because the data augmentation is only applied to a limited number of epochs and training samples. Additionally, the multi-step and single-step AT inherently have a higher basic training time (generate adversarial perturbation), but the extra computational overhead introduced by the DOM framework is relatively consistent. As a result, our approach has a relatively smaller impact on the overall training overhead in these scenarios.

