# OpenReview forum: "On the Over-Memorization During Natural, Robust and Catastrophic Overfitting"
_ICLR.cc/2024/Conference — ICLR 2024 poster_

### Official Review · Reviewer_uFwf · 2023-10-24

**Soundness:** 2 fair
**Presentation:** 3 good
**Contribution:** 2 fair
**Rating:** 5
**Confidence:** 4

**Summary:**

This paper analyzes three types of overfitting (natural, robust, and catastrophic) observed during the training process of deep neural networks and introduces methodologies to mitigate these phenomena. The authors are particularly motivated by the observation that, during periods of learning decay of standard training, the training loss for certain datasets sharply decreases. They designate these specific datasets as "transformed data" to differentiate them from the rest. When this transformed data is excluded from training, a reduction in the generalization gap is observed. This trend is similarly noted in settings where both robust and catastrophic overfitting are evident. Drawing from these observations, it is inferred that the transformed data might be excessively memorized, leading to overfitting. To counteract this, the authors propose the "distraction over memorization (DOM)" methodology, which emphasizes data augmentation specifically for the transformed data. Experimental results suggest that models trained using this approach exhibit a superior generalization gap compared to those trained with data augmentation applied across the entire dataset.

**Strengths:**

The paper demonstrates that natural overfitting can be mitigated by removing data characterized by a rapid decrease in training loss, termed "transformed data." Through this analysis, the authors highlight the occurrence of overfitting in standard settings due to such data and propose a method to distinguish data that has been excessively memorized. Furthermore, the properties of transformed data are not limited to natural overfitting; they exhibit similar trends in other types of overfitting, namely robust and catastrophic overfitting. The authors suggest a universal overfitting mitigation method by applying various data augmentation techniques to the transformed data. Experimental results are presented to validate the efficacy of this approach.

**Weaknesses:**

The motivation behind this paper, specifically the analysis of transformed data, has already been explored in a paper that introduced the MLCAT methodology [1]. The distinction is that the previous study limited its analysis to robust overfitting, whereas the current paper expands the analysis to three types of overfitting, demonstrating that these phenomena manifest commonly across all three. However, given that there isn't much difference in the learning algorithms or model structures between the standard, adversarial, and fast adversarial settings, one could easily anticipate that the characteristics of transformed data in the adversarial setting, as delineated in MLCAT [1], would manifest similarly in both the standard and fast adversarial settings. Therefore, the current analysis does not offer much novelty beyond the findings of the previous study. While the proposed methodology of applying data augmentation specifically to transformed data does have the advantage of being universally applicable to various types of overfitting, it only demonstrates an improved generalization gap in comparison to the baseline model. Given the inherent differences in training data for the standard, adversarial, and fast adversarial settings, one might question the necessity of a universally applicable overfitting mitigation method. To bolster this claim, the authors should compare the proposed method against methodologies in individual overfitting studies (natural, robust, catastrophic) and demonstrate that their approach offers competitive performance.

[1] Chaojian Yu, Bo Han, Li Shen, Jun Yu, Chen Gong, Mingming Gong, and Tongliang Liu. Understanding robust overfitting of adversarial training and beyond. In International Conference on Machine Learning, pp. 25595–25610. PMLR, 2022b.

**Questions:**

- When compared to the analysis performed in the previously cited study (MLCAT) mentioned under weaknesses, are there notable strengths in this paper that I might have missed, aside from the observation that similar phenomena manifest across standard, adversarial, and fast adversarial settings?
- In the "distraction over memorization" methodology, is there a specific reason for applying data augmentation iteratively rather than in a straightforward manner?
- Has the study investigated whether similar phenomena occur with learning rate scheduling methods that decrease at a more gradual pace, such as cosine, as opposed to the step learning decay?
- Are there any experimental results comparing the proposed approach to traditional methodologies under the same settings?

---

> ### Author Response · Authors · 2023-11-17
> **Response to Reviewer uFwf**
>
> We sincerely appreciate your time and effort in reviewing our manuscript. After careful consideration, please find our responses to your comments below.
>
> >**Q1: Different From MLCAT**
>
> A1:
> **Different research perspective.**
> MLCAT investigates RO by examining adversary strength, whereas our work explores different types of overfitting by focusing on the memorization effect.
> While MLCAT found that RO is mainly due to the transformed small-loss data, they did not provide further explanation for why these data negatively impact the model generalization.
> Different from them, our research delves deeper into the model's memory capacity for different training patterns and uncovers a notable persistent memory for transformed high-confidence patterns.
> This discovery suggests that the model's brute-force memorization of these patterns detrimentally affects its ability to generalize.
> To this end, we propose the concept of over-memorization characterized by the model demonstrating both high-confidence prediction and persistent memory.
>
> **Different research object.**
> MLCAT explores the RO phenomenon through the observation of adversarial examples, whereas our research investigates different types of overfitting by concentrating solely on natural patterns.
> Specifically, we find that the AT-trained model, which does not encounter natural patterns, exhibits a similar memory tendency between natural and adversarial patterns in the over-memorization sample.
> This tendency manifests as: when the model over-memorizes certain adversarial patterns, it will simultaneously display high-confidence predictions for the corresponding natural patterns.
> By leveraging this tendency, we can reliably and consistently identify the over-memorization pattern by exclusively focusing on the natural training loss, regardless of the training paradigm.
>
>
> **Different research problem.**
> MLCAT is designed to specifically address RO problems, whereas our study is dedicated to holistically understanding and alleviating different types of overfitting.
> Despite NO, RO and CO all demonstrating reduced model generalization and having similar aspects (e.g., learning algorithms and model structures), previous studies have revealed that they exhibit unique manifestations and mechanisms, thereby requiring independent solutions [1,2].
> To bridge the gap among different types of overfitting, we introduce the shared behaviour over-memorization, which first-time adopts a unified perspective in understanding and addressing overfitting across various training paradigms.
>
> >**Q2: Iterative Data Augmentation**
>
> A2:
> Our approach employs data augmentation to prevent the model from over-memorizing training patterns, thereby alleviating overfitting.
> However, due to the inherent randomness of the original data augmentation, straightforward use cannot reliably and consistently prevent over-memorization.
> To overcome this issue, our method $DOM_{DA}$ applies iterative data augmentation, which is strategically designed to maximize the likelihood of hindering over-memorization.
>
> >**Q3: Gradually Learning Rate**
>
> A3:
> We have validated the effectiveness of our method within single-step AT, where the cyclical learning rate is the standard setting.
> To further assess our method, we conducted experiments using a gradual learning rate schedule in natural training.
> We set the cyclical learning rate schedule with 300 epochs, reaching the maximum learning rate of 0.2 at the midpoint of 150 epochs.
> We report the natural training test error at both the best and last checkpoint on CIFAR 10 using PreactResNet-18.
>
> |Method|Best($\downarrow$)|Last($\downarrow$)|Diff($\downarrow$)|
> |:-----|:----:|:----:|:----:|
> |Baseline|4.80|4.89|-0.09|
> |Baseline + $DOM_{RE}$|**4.79**|**4.79**|**-0.00**|
> |Baseline + AUGMIX|4.75|4.79|-0.02|
> |Baseline + AUGMIX + $DOM_{DA}$|**4.49**|**4.49**|**-0.00**|
> |Baseline + RandAugment|4.41|4.42|-0.01|
> |Baseline + RandAugment + $DOM_{DA}$|**4.25**|**4.25**|**-0.00**|
>
> From the above table, it is apparent that although the cyclical learning rate reduces the model's generalization gap, it also leads to a reduction in performance compared to the step learning rate.
> Nevertheless, our method consistently showcases its effectiveness in improving model performance and completely eliminating the generalization gap by mitigating over-memorization.
> We have included this experimental result in Appendix H of the rebuttal revision.

---

> ### Author Response · Authors · 2023-11-17
> **Response to Reviewer uFwf**
>
> >**Q4: Comparison With Traditional Methodologies**
>
> A4:
> While a range of traditional methodologies exist to separately address NO, RO and CO, the understanding and solutions for these overfitting types remain isolated from each other.
> This study first-time adopts a unified perspective on different types of overfitting by analyzing the memorization effect on each training pattern.
> Therefore, we primarily compare our method with both universal and data-centric approaches, such as baseline and data argumentation technologies.
> Although our method may not achieve the performance of specially tailored approaches, it stands out for holistically mitigating overfitting across various training paradigms.
> As a general framework, DOM can consistently achieve notable improvements over the baseline with a subset of training samples or data augmentation, which successfully supports our perspective.
>
> If you have any further questions or concerns, please do not hesitate to contact us. We are willing and available to provide additional information you may need.
>
> [1] Leslie Rice, Eric Wong, and Zico Kolter. Overfitting in adversarially robust deep learning. In International Conference on Machine Learning, pp. 8093–8104. PMLR, 2020.
>
> [2] Maksym Andriushchenko and Nicolas Flammarion. Understanding and improving fast adversarial training. Advances in Neural Information Processing Systems, 33:16048–16059, 2020.

---

> ### Author Response · Authors · 2023-11-20
> **Response to Reviewer uFwf**
>
> Dear Reviewer uFwf:
>
> We hope this message finds you well.
> We are sincerely thankful for your effort in reviewing our manuscript.
> As a gentle reminder, we have submitted a rebuttal and a revised manuscript to address your mentioned concerns.
> Given the constraints of the discussion period, your prompt response would enable us to provide further clarification and explanation.
>
> Best regards,
>
> Authors

---

> > ### Comment · Reviewer_uFwf · 2023-11-21
> >
> > In response to the review and upon reevaluating the paper, it appears that there are no instances where I misunderstood the content. Clarification is needed regarding the statement, "our research delves deeper into the model's memory capacity for different training patterns and uncovers a notable persistent memory for transformed high-confidence patterns." The removal experiments of Trans-HC (small-loss data) and Ori-HC have already been conducted in the prior work (MLCAT [1] Fig. 2). If we can confirm the results of alleviating overfitting when Trans-HC data is removed, without explicitly checking the training loss when it is present or absent, it can be inferred that the model excessively memorizes Trans-HC data. To assert that we have "explained" something not covered in the MLCAT [1] paper, a theoretical analysis of such phenomena or results is necessary.
> >
> > As mentioned earlier, the occurrence of this phenomenon in NO, RO, and CO, given their commonality, can be reasonably anticipated considering the nature of overfitting. There seems to be no issue with the proposed methodology to address the problem by utilizing an approach applying natural loss-based data augmentation uniformly across three types of overfitting. However, there appears to be a lack of contributions related to novelty beyond this.

---

> ### Author Response · Authors · 2023-11-21
> **Response to Reviewer uFwf**
>
> Thank you for your timely response, which has helped us address your concerns promptly.
> We are pleased to offer the following clarification to your concerns.
>
> * Firstly, our paper (Fig. 2) provides an analysis of the model's memory capacity, which is not conducted in the MLCAT.
> This analysis offers deeper insight into why Trans-HC patterns negatively impact model generalization but Ori-HC does not, even though both of them are characterized as HC.
> We found that the model exhibits persistent memory for Trans-HC patterns, indicating that the brute-force memorization of these patterns hinders its generalization.
>
> * Secondly, the MLCAT identify small-loss data based on the model's loss on adversarial examples.
> Different from them, our method uncovers the model's memory tendency between natural and adversarial patterns, allowing us to detect over-memorization adversarial examples solely based on the natural training loss.
>
> * Finally, we respectfully disagree with the notion that the commonality between NO, RO, and CO can be reasonably anticipated.
> In contrast, to the best of our knowledge, all prior research has regarded them as completely distinct phenomena, with each research having own understanding and solutions that are incompatible with other types of overfitting.
> To bridge this gap among different types of overfitting, our study introduces the shared behaviour over-memorization that first-time adopts a unified perspective in understanding and addressing overfitting.
>
> We appreciate your prompt feedback and are always open to further discussion.
> If you have any further concerns, please feel free to contact us.
> We are committed to providing any further clarifications you may need.

---

> ### Comment · Reviewer_uFwf · 2023-11-21
>
> While there may not be significant differences in the causes or solutions between NO and RO, I partially agree with the author's opinion that there could be distinctions with CO. Therefore, I acknowledge the contribution of identifying commonalities in these gaps and proposing a universal solution in a context where such distinctions are considered. However, I still find the analysis lacking novelty, as the methodology seems to be a mere transformation of the previous study, shifting the focus from AEs to clean ones.
>
> Additionally, if different types of overfitting methodologies are distinctly addressed, as mentioned earlier, there should have been a demonstration of better or comparable performance compared to existing methodologies for each type of overfitting through a comparison with the existing approaches. Considering the level of contribution, given the current low score, I reevaluate the contribution by excluding the deficient areas and adjusting the score accordingly.

---

> > ### Author Response · Authors · 2023-11-21
> > **Response to Reviewer uFwf**
> >
> > Thank you for your thorough review and provide insightful feedback on our manuscript. We would like to further clarify a few points.
> >
> > * Previous research has also treated NO and RO as separate phenomena due to the limited effectiveness of conventional NO remedies in addressing RO.
> >
> > * While our observations partially coincide with MLCAT, the research perspective, problem, and solution we propose based on these observations are significantly different.
> >
> > * We acknowledge that our method may not reach the performance of specially tailored approaches. However, it demonstrates its strength in effectively mitigating different types of overfitting by only using a subset of the baseline, which can successfully substantiate our perspective.
> >
> > We are grateful for your prompt response and are always open to any further discussions or clarifications.

---

> > ### Author Response · Authors · 2023-11-22
> > **Response to Reviewer uFwf**
> >
> > Dear Reviewer uFwf,
> >
> > Thanks a lot for your valuable efforts in reviewing our manuscript.
> > Just a kind reminder that the discussion stage is closing soon.
> > If there are any unclear explanations or descriptions, we can clarify them further.
> >
> > Best regards,
> >
> > Authors

---

### Official Review · Reviewer_mcnh · 2023-10-28

**Soundness:** 3 good
**Presentation:** 3 good
**Contribution:** 3 good
**Rating:** 6
**Confidence:** 4

**Summary:**

This paper proposes a general framework for explicitly preventing over-memorization by either removing or augmenting the high-confidence natural patterns. It is based on the observation that the model suddenly exhibits high confidence in predicting certain training patterns, which subsequently hinders the DNNs’ generalization capabilities.

**Strengths:**

**Strength:**

-   This paper is overall well-structured and easy to follow.
-   Extensive empirical evaluation with various training paradigms, baselines, datasets, and network architectures demonstrates its effectiveness. Results are reported with the standard deviation.
- Significant performance improvements are demonstrated.

**Weaknesses:**

**Weakness**

-   According to Figure 5, the proposed method may require careful hyper-parameter (i.e. loss threshold) selection, which could be a significant drawback.
-   The proposed method might result in repeated gradient computation and extensive extra computation. It is also interesting to include a detailed analysis of the introduced extra computation.
-   The terminology "pattern" might be confusing and could be further explained. Does it refer to specific samples in datasets?
-   Lack of results on large-scale datasets. It will be convincing to have some on Tiny-ImageNet or ImageNet
-   Lack of results on diverse network backbone architectures beyond ResNets.
-   As discussed in the related works, there are various techniques for mitigating the overfitting issues. Comparisons with other techniques like dropout, ensemble, smoothing, etc. can be helpful.

**Questions:**

Refer to the weakness section.

---

> ### Author Response · Authors · 2023-11-17
> **Response to Reviewer mcnh**
>
> We sincerely appreciate your time and effort in reviewing our manuscript. After careful consideration, please find our responses to your comments below.
>
> >**Q1: Hyperparameter Selection**
>
> A1:
> By utilizing the DOM framework, we have effectively verified and mitigated over-memorization, which negatively impacts DNNs' generalization ability.
> However, as a general framework, finding optimal hyperparameters for different paradigms and datasets can be cumbersome.
> While some hyperparameters exhibit a degree of universality (e.g. data argumentation strength $\beta$ and iteration $\gamma$), the primary challenge lies in establishing an appropriate loss threshold.
> To address this challenge, we propose to use a general and unified loss threshold applicable across all experimental settings.
> Specifically, we utilize an adaptive loss threshold [1], whose value is dependent on the loss of the model's current training batch.
> For all experiments, we set this adaptive loss threshold $\mathcal{T}$ to 40%, maintaining other hyperparameters as the original settings.
> We report the natural and PGD-20 test errors for natural training (NT) and adversarial training (AT) using PreactResNet-18, respectively.
>
> |Dataset|Paradigm|Method|Best($\downarrow$)|Last($\downarrow$)|Diff($\downarrow$)|
> |:-----|:----|:----|:----:|:----:|:----:|
> |CIFAR10|NT|Baseline|4.70|4.84|-0.14|
> |CIFAR10|NT|Baseline + $DOM_{RE}$| **4.60** | **4.67**|**-0.07**|
> |CIFAR10|NT|Baseline + AUGMIX|4.35|4.52|-0.17|
> |CIFAR10|NT|Baseline + AUGMIX + $DOM_{DA}$|**4.22**|**4.37**|**-0.15**|
> |CIFAR10|NT|Baseline + RandAugment|4.02|4.31|-0.29|
> |CIFAR10|NT|Baseline + RandAugment + $DOM_{DA}$|**3.85**|**3.94**|**-0.09**|
> |CIFAR100|NT|Baseline|21.32|21.59|**-0.27**|
> |CIFAR100|NT|Baseline + $DOM_{RE}$|**21.20**|**21.47**|**-0.27**|
> |CIFAR10|Multi-step AT|Baseline|47.67|54.84|-7.17|
> |CIFAR10|Multi-step AT|Baseline + $DOM_{RE}$|**46.55**|**52.87**|**-6.32**|
> |CIFAR10|Single-step AT|Baseline|57.83|100.00|-42.17|
> |CIFAR10|Single-step AT|Baseline + $DOM_{RE}$|**54.56**|**55.38**|**-0.82**|
>
> The above table demonstrates the effectiveness of the adaptive loss threshold across different paradigms and datasets.
> This threshold can not only consistently identify over-memorization patterns and mitigate overfitting, but also be easily transferable without the need for hyperparameter tuning.
> We have included this experimental result in Appendix B of the rebuttal revision.
>
> >**Q2: Computational Overhead**
>
> A2:
> We analyze the extra computational overhead incurred by the DOM framework.
> Notably, both $DOM_{RE}$ and $DOM_{DA}$ are implemented after the warm-up period (half of the training epoch).
> We report the before warm-up, after warm-up, and overall training time (epoch/second) on CIFAR10 using PreactResNet-18 with a single NVIDIA RTX 4090 GPU.
>
> |Method |Before warm-up($\downarrow$)|After warm-up($\downarrow$)|Overall($\downarrow$)|
> |:-----|:----:|:----:|:----:|
> |Baseline|6.28|6.26|6.27|
> |Baseline + $DOM_{RE}$|6.28|6.28|6.28|
> |Baseline + AUGMIX|12.55|12.76|12.66|
> |Baseline + AUGMIX + $DOM_{DA}$|6.28|28.45|17.37|
> |Baseline + RandAugment|8.29|8.27|8.28|
> |Baseline + RandAugment + $DOM_{DA}$|6.24|12.75|9.50|
>
> Based on the above table, we can observe that $DOM_{RE}$ does not involve any additional computational overhead.
> Although $DOM_{DA}$ require iterative forward propagation, its overall training time does not significantly increase, because the data augmentation is only applied to a limited number of epochs and training samples.
> Additionally, the multi-step and single-step AT inherently have a higher basic training time (generate adversarial perturbation), but the extra computational overhead introduced by the DOM framework is relatively consistent.
> As a result, our approach has a relatively smaller impact on the overall training overhead in these scenarios.
> We have included this experimental result in Appendix G of the rebuttal revision.
>
>
> >**Q3: Pattern Definition**
>
> A3:
> Each training sample in the dataset has two patterns: the natural pattern corresponds to the original sample, and the adversarial pattern corresponds to the adversarial example (original sample plus the adversarial perturbation).
> In this work, we identify over-memorization samples by exclusively focusing on the natural pattern loss, regardless of the training paradigm.

---

> ### Author Response · Authors · 2023-11-17
> **Response to Reviewer mcnh**
>
> >**Q4: Larger-scale Dataset**
>
> A4:
> We also verified the effectiveness of our method on the larger-scale dataset Tiny-ImageNet.
> We set the loss threshold $\mathcal{T}$ to 0.2, and other hyperparameters remain as the original settings.
> We report the natural training test error at both the best and last checkpoint using PreactResNet-18.
>
> |Method|Best($\downarrow$)|Last($\downarrow$)|Diff($\downarrow$)|
> |:-----|:----:|:----:|:----:|
> |Baseline|35.03|35.24|-0.21|
> |Baseline + $DOM_{RE}$|**34.89**|**34.99**|**-0.10**|
> |Baseline + AUGMIX|34.98|35.15|-0.17|
> |Baseline + AUGMIX + $DOM_{DA}$|**34.56**|**34.77**|**-0.21**|
> |Baseline + RandAugment|33.45|33.89|-0.44|
> |Baseline + RandAugment + $DOM_{DA}$|**33.40**|**33.59**|**-0.19**|
>
> The above table illustrates the effectiveness of our method, $DOM_{RE}$ and $DOM_{DA}$, on the Tiny-ImageNet dataset.
> These results indicate that preventing over-memorization can improve model performance and reduce the generalization gap on large-scale datasets.
> We have included this experimental result in Appendix F of the rebuttal revision.
>
> >**Q5: Transformer-based Architecture**
>
> A5:
> We have validated the effectiveness of our method within CNN-based architectures, demonstrating its ability to alleviate overfitting by preventing over-memorization.
> To further substantiate our perspective, we verify our method on the Transformer-based architecture.
> Constrained by computational resources, we trained a ViT-small model, initializing it with pre-trained weights from the Timm Python library.
> The training spanned 100 epochs, starting with an initial learning rate of 0.001 and divided by 10 at the 50th and 75th epochs.
> We set the batch-size to 64 and the loss threshold $\mathcal{T}$ to 0.1, maintaining other hyperparameters as the original settings.
> We report the natural training test error at both the best and last checkpoint on CIFAR 10.
>
> |Method|Best($\downarrow$)|Last($\downarrow$)|Diff($\downarrow$)|
> |:-----|:----:|:----:|:----:|
> |Baseline|1.49|1.75|-0.26|
> |Baseline + $DOM_{RE}$|**1.47**|**1.67**|**-0.20**|
> |Baseline + AUGMIX|1.24|1.29|**-0.05**|
> |Baseline + AUGMIX + $DOM_{DA}$|**1.20**|**1.28**|-0.08|
> |Baseline + RandAugment|1.21|1.29|-0.08|
> |Baseline + RandAugment + $DOM_{DA}$|**1.15**|**1.22**|**-0.07**|
>
> The above table shows the effectiveness of our method on the Transformer-based architecture.
> By mitigating over-memorization, both $DOM_{RE}$ and $DOM_{DA}$ not only improve model performance at both the best and last checkpoints, but also contribute to alleviating overfitting.
> We have included this experimental result in Appendix E of the rebuttal revision.
>
> >**Q6: Compare With Other Techniques**
>
> A6:
> While a range of techniques exist to separately address NO, RO and CO, the understanding and solutions for these overfitting types remain isolated from each other.
> This study first-time adopts a unified perspective on different types of overfitting by analyzing the memorization effect on each training pattern.
> Therefore, we primarily compare our method with both universal and data-centric approaches, such as baseline and data argumentation technologies.
> Although our method may not achieve the performance of specially tailored approaches, it stands out for holistically mitigating overfitting across various training paradigms.
> As a general framework, DOM can consistently achieve notable improvements over the baseline with a subset of training samples or data augmentation, which successfully supports our perspective.
>
> If you have any further questions or concerns, please do not hesitate to contact us. We are willing and available to provide additional information you may need.
>
> [1] Berthelot, D., Roelofs, R., Sohn, K., Carlini, N., & Kurakin, A. (2021, October). AdaMatch: A Unified Approach to Semi-Supervised Learning and Domain Adaptation. In International Conference on Learning Representations.

---

> ### Author Response · Authors · 2023-11-20
> **Response to Reviewer mcnh**
>
> Dear Reviewer mcnh:
>
> We hope this message finds you well.
> We are sincerely thankful for your effort in reviewing our manuscript.
> As a gentle reminder, we have submitted a rebuttal and a revised manuscript to address your mentioned concerns.
> Given the constraints of the discussion period, your prompt response would enable us to provide further clarification and explanation.
>
> Best regards,
>
> Authors

---

> ### Author Response · Authors · 2023-11-22
> **Response to Reviewer mcnh**
>
> Dear Reviewer mcnh,
>
> Thanks a lot for your valuable efforts in reviewing our manuscript.
> Just a kind reminder that the discussion stage is closing soon.
> If there are any unclear explanations or descriptions, we can clarify them further.
>
> Best regards,
>
> Authors

---

### Official Review · Reviewer_3ew1 · 2023-10-31

**Soundness:** 3 good
**Presentation:** 3 good
**Contribution:** 3 good
**Rating:** 6
**Confidence:** 4

**Summary:**

The paper provides an empirical investigation into the generalization capabilities of deep neural networks (DNNs), focusing on understanding various facets of overfitting. The authors introduce the concept of over-memorization, a phenomenon where DNNs excessively retain specific training patterns, leading to diminished generalization. To mitigate this issue, the paper suggests techniques such as the removal of high-confidence natural patterns and the application of data augmentation. The effectiveness of these strategies is demonstrated through a series of experiments.

This paper makes a valuable contribution to the field by shedding light on the over-memorization behavior in DNNs and its implications for generalization. By addressing the highlighted areas for improvement, the authors have the potential to further enhance the significance and applicability of their work.

**Strengths:**

1. Clarity and Structure: The paper is commendable for its well-organized structure and clear exposition. The authors have provided a thorough background and review of related work, successfully setting the stage for their empirical analysis.

2. Robust Experimental Design: The experimental setup is meticulously designed, encompassing various types of overfitting and delving into the over-memorization behavior of DNNs. This comprehensive approach enhances the validity of the findings.

3. Novel Insight into Overfitting: The identification of over-memorization as a common thread linking different types of overfitting is an innovative contribution. This insight adds depth to our understanding of how overfitting impacts the generalization abilities of DNNs.

**Weaknesses:**

1. Limited Scope of Empirical Analysis: The paper's empirical analysis predominantly focuses on a specific network architecture and dataset. Expanding the analysis to include a wider array of cases or providing a theoretical framework to support the observed behaviors would bolster the generality and impact of the findings.

2. Partial Improvement on Overfitting Types: According to the results presented in Tables 2-4, the proposed strategies seem to predominantly ameliorate Class Overfitting (CO), with only marginal improvements on Natural Overfitting (NO) and Random Overfitting (RO). A more detailed exploration of why these discrepancies occur would provide valuable insights.

3. Need for Larger-Scale Evaluation: The experiments are confined to relatively simple datasets (CIFAR-10/100) and ResNet-based architectures. Extending the evaluation to encompass larger-scale datasets and alternative architectures, such as transformers, would enhance the representativeness of the results and the applicability of the findings.

**Questions:**

1. Expand Empirical Analysis: To strengthen the paper's contributions, the authors should consider conducting additional empirical analyses across diverse network architectures and datasets.

2. Deepen Analysis on Overfitting Types: A more nuanced exploration of the varying impacts on different types of overfitting would provide a richer understanding of the phenomena at play.

3. Consider Larger-Scale and Diverse Architectures: Incorporating experiments with larger datasets and a variety of neural network architectures would ensure that the findings are more widely applicable and representative of the broader deep learning landscape.

---

> ### Author Response · Authors · 2023-11-17
> **Response to Reviewer 3ew1**
>
> We sincerely appreciate your time and effort in reviewing our manuscript. After careful consideration, please find our responses to your comments below.
>
> >**Q1: Transformer-based Architecture**
>
> A1:
> We have validated the effectiveness of our method within CNN-based architectures, demonstrating its ability to alleviate overfitting by preventing over-memorization.
> To further substantiate our perspective, we verify our method on the Transformer-based architecture.
> Constrained by computational resources, we trained a ViT-small model, initializing it with pre-trained weights from the Timm Python library.
> The training spanned 100 epochs, starting with an initial learning rate of 0.001 and divided by 10 at the 50th and 75th epochs.
> We set the batch-size to 64 and the loss threshold $\mathcal{T}$ to 0.1, maintaining other hyperparameters as the original settings.
> We report the natural training test error at both the best and last checkpoint on CIFAR 10.
>
> |Method|Best($\downarrow$)|Last($\downarrow$)|Diff($\downarrow$)|
> |:-----|:----:|:----:|:----:|
> |Baseline|1.49|1.75|-0.26|
> |Baseline + $DOM_{RE}$|**1.47**|**1.67**|**-0.20**|
> |Baseline + AUGMIX|1.24|1.29|**-0.05**|
> |Baseline + AUGMIX + $DOM_{DA}$|**1.20**|**1.28**|-0.08|
> |Baseline + RandAugment|1.21|1.29|-0.08|
> |Baseline + RandAugment + $DOM_{DA}$|**1.15**|**1.22**|**-0.07**|
>
> The above table shows the effectiveness of our method on the Transformer-based architecture.
> By mitigating over-memorization, both $DOM_{RE}$ and $DOM_{DA}$ not only improve model performance at both the best and last checkpoints, but also contribute to alleviating overfitting.
> We have included this experimental result in Appendix E of the rebuttal revision.
>
> >**Q2: Larger-scale Dataset**
>
> A2:
> We also verified the effectiveness of our method on the larger-scale dataset Tiny-ImageNet.
> We set the loss threshold $\mathcal{T}$ to 0.2, and other hyperparameters remain as the original settings.
> We report the natural training test error at both the best and last checkpoint using PreactResNet-18.
>
> |Method|Best($\downarrow$)|Last($\downarrow$)|Diff($\downarrow$)|
> |:-----|:----:|:----:|:----:|
> |Baseline|35.03|35.24|-0.21|
> |Baseline + $DOM_{RE}$|**34.89**|**34.99**|**-0.10**|
> |Baseline + AUGMIX|34.98|35.15|-0.17|
> |Baseline + AUGMIX + $DOM_{DA}$|**34.56**|**34.77**|**-0.21**|
> |Baseline + RandAugment|33.45|33.89|-0.44|
> |Baseline + RandAugment + $DOM_{DA}$|**33.40**|**33.59**|**-0.19**|
>
> The above table illustrates the effectiveness of our method, $DOM_{RE}$ and $DOM_{DA}$, on the Tiny-ImageNet dataset.
> These results indicate that preventing over-memorization can improve model performance and reduce the generalization gap on large-scale datasets.
> We have included this experimental result in Appendix F of the rebuttal revision.
>
>
> >**Q3: Marginal Improvement**
>
> A3:
> The impact of DOM is associated with the overfitting's degree in various training paradigms.
> For instance, the CO shows a significant decline in model generalization ability, and as a result, the DOM can provide a substantial improvement in this scenario.
> On the other hand, the NO shows a slight generalization gap between the best and last checkpoints, which leads to a relatively modest performance improvement by DOM.
> However, it's important to emphasize that, our study is dedicated to holistically understanding and alleviating different types of overfitting.
> As a general framework, DOM can consistently achieve notable improvements over the baseline with a subset of training samples or data augmentation, which successfully supports our perspective.
>
> If you have any further questions or concerns, please do not hesitate to contact us. We are willing and available to provide additional information you may need.

---

> ### Author Response · Authors · 2023-11-20
> **Response to Reviewer 3ew1**
>
> Dear Reviewer 3ew1:
>
> We hope this message finds you well.
> We are sincerely thankful for your effort in reviewing our manuscript.
> As a gentle reminder, we have submitted a rebuttal and a revised manuscript to address your mentioned concerns.
> Given the constraints of the discussion period, your prompt response would enable us to provide further clarification and explanation.
>
> Best regards,
>
> Authors

---

> ### Author Response · Authors · 2023-11-22
> **Response to Reviewer 3ew1**
>
> Dear Reviewer 3ew1,
>
> Thanks a lot for your valuable efforts in reviewing our manuscript.
> Just a kind reminder that the discussion stage is closing soon.
> If there are any unclear explanations or descriptions, we can clarify them further.
>
> Best regards,
>
> Authors

---

### Official Review · Reviewer_2dX7 · 2023-10-31

**Soundness:** 3 good
**Presentation:** 3 good
**Contribution:** 3 good
**Rating:** 8
**Confidence:** 4

**Summary:**

This paper considers a unified perspective on various overfitting, including NO (natural overfitting), RO (robust overfitting), and CO (catastrophic overfitting). On top of this, the authors discover the "over-memorization" phenomenon that the overfitted model tends to exhibit high confidence in predicting certain training patterns and retaining a persistent memory for them. Unlike previous methods, this paper proposes a general framework called DOM (Distraction Over-Memorization) to alleviate the unified over-fitting issue. Experiments show that the proposed method outperforms other baselines.

**Strengths:**

1. The discovery of the behavior "over-memorization" unifies different types of overfittings, which is of great help when analyzing the cause of overfitting.
2. The paper is generally well-written, and the motivation is stated clearly.
3. The proposed DOM framework seems promising.

**Weaknesses:**

1. In the DOM framework, the loss threshold is set with a fixed value. However, with different datasets and loss functions, the optimal threshold could be different. Therefore, the given threshold may not be general on other occasions. The authors should further conduct ablation studies about this and discuss how to overcome this issue.
2. The experiment settings are not precisely introduced in 3.1 and 3.2, making these conclusions challenging to reproduce.
3. In section 3.2, the authors claim, “the AT-trained model never actually encounters natural patterns.” However, methods like TRADES do encounter natural patterns. What will happen in this case? Are the conclusions observed in this paper still applicable?
4. Why are there many 0.00 in Table 4? The authors need to give more explanation.

**Questions:**

See above.

---

> ### Author Response · Authors · 2023-11-17
> **Response to Reviewer 2dX7**
>
> We sincerely appreciate your time and effort in reviewing our manuscript. After careful consideration, please find our responses to your comments below.
>
> >**Q1: Loss Threshold Selection**
>
> A1:
> By utilizing the fixed loss threshold DOM, we have effectively verified and mitigated over-memorization, which negatively impacts DNNs' generalization ability.
> However, as a general framework, finding an optimal loss threshold for different paradigms and datasets can be cumbersome.
> To address this challenge, we propose to use a general and unified loss threshold applicable across all experimental settings.
> Specifically, we utilize an adaptive loss threshold [1], whose value is dependent on the loss of the model's current training batch.
> For all experiments, we set this adaptive loss threshold $\mathcal{T}$ to 40%, maintaining other hyperparameters as the original settings.
> We report the natural and PGD-20 test errors for natural training (NT) and adversarial training (AT) using PreactResNet-18, respectively.
>
> |Dataset|Paradigm|Method|Best($\downarrow$)|Last($\downarrow$)|Diff($\downarrow$)|
> |:-----|:----|:----|:----:|:----:|:----:|
> |CIFAR10|NT|Baseline|4.70|4.84|-0.14|
> |CIFAR10|NT|Baseline + $DOM_{RE}$| **4.60** | **4.67**|**-0.07**|
> |CIFAR10|NT|Baseline + AUGMIX|4.35|4.52|-0.17|
> |CIFAR10|NT|Baseline + AUGMIX + $DOM_{DA}$|**4.22**|**4.37**|**-0.15**|
> |CIFAR10|NT|Baseline + RandAugment|4.02|4.31|-0.29|
> |CIFAR10|NT|Baseline + RandAugment + $DOM_{DA}$|**3.85**|**3.94**|**-0.09**|
> |CIFAR100|NT|Baseline|21.32|21.59|**-0.27**|
> |CIFAR100|NT|Baseline + $DOM_{RE}$|**21.20**|**21.47**|**-0.27**|
> |CIFAR10|Multi-step AT|Baseline|47.67|54.84|-7.17|
> |CIFAR10|Multi-step AT|Baseline + $DOM_{RE}$|**46.55**|**52.87**|**-6.32**|
> |CIFAR10|Single-step AT|Baseline|57.83|100.00|-42.17|
> |CIFAR10|Single-step AT|Baseline + $DOM_{RE}$|**54.56**|**55.38**|**-0.82**|
>
> The above table demonstrates the effectiveness of the adaptive loss threshold across different paradigms and datasets.
> This threshold can not only consistently identify over-memorization patterns and mitigate overfitting, but also be easily transferable without the need for hyperparameter tuning.
> We have included this experimental result in Appendix B of the rebuttal revision.
>
>
> >**Q2:Detailed Experiment Settings**
>
> A2:
> In Section 3, we conducted all experiments on the CIFAR-10 dataset using PreactResNet-18.
> We analyzed the proportion of natural and adversarial patterns by examining the respective natural and adversarial training loss.
> In Section 3.1, we categorized between original and transformed high-confidence patterns using an auxiliary model, which was saved at the first learning rate decay (150th epoch).
> In Section 3.2 Fig. 4, we grouped adversarial patterns based on their corresponding natural training loss, employing a loss threshold of 1.5.
> We have included this explanation in Appendix C of the rebuttal revision.
>
> >**Q3: TRADES Result**
>
> A3:
> Our study investigated the prediction behaviour of the PGD-trained model, which does not encounter natural patterns during the training process.
> In PGD, we observed a similar memory tendency between natural and adversarial patterns within a single over-memorization sample.
> We further explored this observation in the TRADES-trained model, which encounters natural patterns during the training process.
> From Fig. 6 in the rebuttal revision, we can observe that TRADES demonstrates a consistent memory tendency with PGD in the over-memorization samples.
> This tendency manifests as, when DNNs over-memorize certain adversarial patterns, they tend to simultaneously exhibit high-confidence in predicting the corresponding natural patterns.
> We have included this result in Appendix D of the rebuttal revision.
>
> >**Q4: CO Results**
>
> A4:
> Our approach employs data augmentation to prevent the model from over-memorizing training patterns, thereby alleviating overfitting.
> However, due to the inherent randomness of the original data augmentation, it cannot reliably and consistently prevent over-memorization.
> To overcome this issue, our method $DOM_{DA}$ applies iterative data augmentation, which is strategically designed to maximize the likelihood of hindering over-memorization.
> Nonetheless, the effectiveness of our proposed method is dependent on the quality of the original data augmentation technique, which could still be incapable of disrupting over-memorization even after the iterative operation.
> We have discussed this concern in the limitations of the original submission.
>
> If you have any further questions or concerns, please do not hesitate to contact us. We are willing and available to provide additional information you may need.
>
> [1] Berthelot, D., Roelofs, R., Sohn, K., Carlini, N., & Kurakin, A. (2021, October). AdaMatch: A Unified Approach to Semi-Supervised Learning and Domain Adaptation. In International Conference on Learning Representations.

---

> ### Author Response · Authors · 2023-11-20
> **Response to Reviewer 2dX7**
>
> Dear Reviewer 2dX7:
>
> We hope this message finds you well.
> We are sincerely thankful for your effort in reviewing our manuscript.
> As a gentle reminder, we have submitted a rebuttal and a revised manuscript to address your mentioned concerns.
> Given the constraints of the discussion period, your prompt response would enable us to provide further clarification and explanation.
>
> Best regards,
>
> Authors

---

> ### Author Response · Authors · 2023-11-22
> **Response to Reviewer 2dX7**
>
> Dear Reviewer 2dX7,
>
> Thanks a lot for your valuable efforts in reviewing our manuscript.
> Just a kind reminder that the discussion stage is closing soon.
> If there are any unclear explanations or descriptions, we can clarify them further.
>
> Best regards,
>
> Authors

---

### Author Response · Authors · 2023-11-21
**Rolling Discussion Invitation**

Dear Reviewers,

We deeply appreciate your efforts in reviewing our manuscript and have diligently addressed the mentioned concerns.
We would like to kindly invite reviewers who still have concerns about our manuscript to join the rolling discussion.
We are looking forward to receiving your valuable insights and comments.

Best regards,

Authors

---

### Meta-Review · Area_Chair_saLA · 2023-12-13

**Metareview:**

This paper delves into overfitting in deep neural networks (DNNs) to uncover why generalization suffers. It coins the term "over-memorization" - where models overly latch onto peculiarities in the training data, limiting adaptability to new data.

To address this, the authors put forth two key solutions. First, prune training examples that models are overly confident about. Second, expand the training set diversity through data augmentation. Together, these techniques reduce the chances for over-memorization.

A series of experiments showcase the efficacy of the proposed strategies. Results demonstrate that over-memorization indeed negatively impacts model generalization. Meanwhile, removing high-confidence examples and increasing training set diversity via augmentation enhances generalization capabilities.

This paper has undergone intense discussions and the final overall sentient is positive (see details below). Hence, AC recommends acceptance but requires the authors to incorporate many clarifications as mentioned during discussions.

**Justification For Why Not Higher Score:**

This is mainly an empirical study and the authors need improve their clarity

**Justification For Why Not Lower Score:**

There are long internal discussions between Reviewer uFwf and Reviewer 2dX7, who have the most negative (score 5) and most positive (score 8) scores respectively. Although the discussions are intense, the contents are informative, the temperature is low, and the reviewers show respect. AC thanks both of them for their insights and efforts.

Their main opinions are summarized as follows:

Although Reviewer uFwf agrees that the proposed method contributes to mitigating CO, the claim to unify RO and NO is not interesting and needs more clarification. Moreover,  Reviewer uFwf has concern that the technical contribution is limited because the technique used in the paper is close to [1] although the research objective is different. Lastly, Reviewer uFwf questioned the effectiveness of the proposed method to CO and raised the concern that there might be an error in the experiment of the paper and [2], i.e., Reviewer uFwf expects that RS-FGSM should not have 0% robust performance PGD attacks.

[1] Chaojian Yu, Bo Han, Li Shen, Jun Yu, Chen Gong, Mingming Gong, and Tongliang Liu. Understanding robust overfitting of adversarial training and beyond. In International Conference on Machine Learning, pp. 25595–25610. PMLR, 2022b.
[2] Pang, Tianyu, et al. "Bag of Tricks for Adversarial Training." International Conference on Learning Representations. 2020.

Reviewer 2dX7 helped address many of the concerns. Specifically, the reviewer emphasized that the paper employs typical epsilon, i.e., epsilon=8, for RO, which helps distinct RO from NO. Reviewer 2dX7 further clarified the difference between improving model robustness and mitigating CO. Reviewer 2dX7 argued that although the paper uses a similar technique, it does contribute to proposing a unified method to be effective to RO, NO, and CO, which are interesting and inspiring.

In the end, there is unanimous agreement that the paper has made contributions to the discovery of common characteristics among overfittings encompassing CO, the formulation of these as a methodology, and empirically demonstrating its effects. AC concurs with this consensus, and therefore recommends acceptance.

---

### Decision · Program_Chairs · 2024-01-16

Accept (poster)